An improved hippopotamus optimization algorithm based on adaptive development and solution diversity enhancement

http://orcid.org/0000-0003-0924-3079 Pei Shengyu 1 2 shengypei@gmail.com
Sun Gang 3
Tong Lang 4
1 School of Artiffcal Intelligence, Guangxi Minzu University , Nanning, Guangxi , China
2 Guangxi Key Laboratory of Hybrid Computation and IC Design Analysis, Guangxi Minzu University , Nanning, Guangxi , China
3 Hunan Tobacco Workers Training Center , Xiangtan, Hunan , China
4 School of Foreign Languages for Business, Guangxi University of Finance and Economics , Nanning, Guangxi , China
Alatas Bilal
Electronic publication date: 2025 May 29
Publication date: 2025
Volume: 11
Electronic Location ID: e2901
Received 2024 Nov 19; Accepted 2025 Apr 25
Copyright: © 2025 Pei et al.
Copyright year: 2025
Copyright holder: Pei et al.
License: This is an open access article distributed under the terms of the Creative Commons Attribution License, which permits unrestricted use, distribution, reproduction and adaptation in any medium and for any purpose provided that it is properly attributed. For attribution, the original author(s), title, publication source (PeerJ Computer Science) and either DOI or URL of the article must be cited.
License URL: https://creativecommons.org/licenses/by/4.0/

Keywords: Hippopotamus optimization algorithm, Adaptive exploitation, Solution diversity, Chaotic mapping, Global optimization

Funding: Research Project of Guangxi Minzu University 2022KJQD020 This work was supported by the Research Project of Guangxi Minzu University (2022KJQD020). The funders had no role in study design, data collection and analysis, decision to publish, or preparation of the manuscript.

==============================
This study proposes an improved hippopotamus optimization algorithm to address the limitations of the traditional hippopotamus optimization algorithm in terms of convergence performance and solution diversity in complex high-dimensional problems. Inspired by the natural behavior of hippopotamuses, this article introduces chaotic map initialization, an adaptive exploitation mechanism, and a solution diversity enhancement strategy based on the original algorithm. The chaotic map is employed to optimize the initial population distribution, thereby enhancing the global search capability. The adaptive exploitation mechanism dynamically adjusts the weights between the exploration and exploitation phases to balance global and local searches. The solution diversity enhancement is achieved through the introduction of nonlinear perturbations, which help the algorithm avoid being trapped in local optima. The proposed algorithm is validated on several standard benchmark functions (CEC17, CEC22), and the results demonstrate that the improved algorithm significantly outperforms the original hippopotamus optimization algorithm and other mainstream optimization algorithms in terms of convergence speed, solution accuracy, and global search ability. Moreover, statistical analysis further confirms the superiority of the improved algorithm in balancing exploration and exploitation, particularly when dealing with high-dimensional multimodal functions. This study provides new insights and enhancement strategies for the application of the hippopotamus optimization algorithm in solving complex optimization problems.

Introduction

With the continuous advancements in science, industry, and technology, many problems are defined as optimization problems (Chen et al., 2024a). These problems typically consist of three fundamental components: an objective function, constraints, and decision variables (Zhang et al., 2022). To address such challenges, optimization algorithms can be classified into various categories. One common classification distinguishes between stochastic and deterministic algorithms based on their inherent optimization approach. Unlike deterministic methods, stochastic approaches do not require comprehensive knowledge of the problem’s characteristics, making them advantageous when dealing with complex, high-dimensional, nonlinear, and non-differentiable problems. Stochastic methods are especially effective when the problem is poorly understood or treated as a black box (Ju & Liu, 2024).

Among the numerous stochastic methods, metaheuristic algorithms have garnered significant attention due to their exceptional performance in solving complex problems (Dai & Fu, 2023; Dong & Chen, 2023). These algorithms generate an initial set of candidate solutions randomly and iteratively update these solutions according to specific relationships defined by the algorithm. In each iteration, better solutions are retained based on the number of search agents until a termination criterion, such as a predefined maximum number of iterations or the number of function evaluations, is met. The advantage of metaheuristic algorithms lies in their ability to balance global and local search, allowing them to excel in various applications (Guo, 2023; Chen, 2023; Zhang, 2023).

The hippopotamus optimization algorithm (HO) is a nature-inspired metaheuristic optimization algorithm first introduced in 2024 by Amiri et al. (2024). The HO draws inspiration from three behavioral patterns observed in hippopotamuses: position updates in water, defensive strategies against predators, and predator evasion. These behaviors are mathematically modeled to guide the optimization process. Although many optimization algorithms have been proposed, the “No Free Lunch” (NFL) theorem suggests that no single algorithm can outperform all others in solving every optimization problem. As increasingly complex optimization problems arise, traditional algorithms often struggle to handle issues such as nonlinearity, non-convexity, and non-differentiability. The HO was designed to balance global exploration and local exploitation by simulating hippopotamus behavior, thereby enhancing convergence speed and solution accuracy in multi-dimensional, multi-modal optimization problems. Despite the HO’s strong performance in benchmark tests and practical applications, it still faces certain limitations common to metaheuristic algorithms (Mashru et al., 2024). Therefore, developing novel and efficient optimization algorithms remains a critical research focus (Trojovsky & Dehghani, 2022).

Metaheuristic algorithms have found widespread applications across various engineering fields, including hyperparameter tuning and neural network weight optimization in medical engineering, intelligent fault diagnosis in control engineering, controller parameter optimization in mechanical engineering, and filter design in telecommunications engineering (Liao et al., 2024; Dehghani & Trojovsky, 2021; Emami, 2022; Trojovsky & Dehghani, 2022; Chen, Niu & Zhang, 2022; Yu et al., 2023; Han et al., 2024). These algorithms have also proven valuable in energy, civil engineering, and economics.

Recent studies have shown that chaotic mapping can significantly enhance the performance of metaheuristic algorithms. The mountain gazelle optimizer (MGO) stands out for its fast convergence and high precision, but it suffers from premature convergence and is prone to being trapped in local optima. Sarangi & Mohapatra (2024) proposed the chaotic mountain gazelle optimizer (CMGO), which leverages multiple chaotic maps to overcome these limitations. The Harris hawks optimization (HHO) algorithm, a novel swarm-based nature-inspired algorithm, has exhibited excellent performance but still has the drawbacks of premature convergence and falling into local optima due to the imbalance between exploration and exploitation. Yang et al. (2023) proposed the HHO-cs-oelm algorithm, which enhances global search capability through chaotic sequences and strengthens local search capability via opposition-based elite learning, thereby balancing exploration and exploitation. The HHO, inspired by the unique foraging strategies and cooperative behavior of Harris hawks, is also prone to local optima and slow convergence. Almotairi et al. (2023) presented several techniques to enhance the performance of metaheuristic algorithms (MHAs) and address their limitations. Chaotic optimization strategies, which have been proposed for many years to enhance MHAs, include four different types: chaotic mapping initialization, stochasticity, iteration, and control parameters. This article introduces a novel hybrid algorithm, SHHOIRC, designed to improve the efficiency of HHO. An adaptive HHO algorithm based on three chaotic optimization methods has been proposed.

To address the shortcomings of existing algorithms in terms of convergence speed and solution accuracy, this article proposes an improved hippopotamus optimization algorithm, hereafter referred to as IHO. The IHO optimizes the process by simulating hippopotamus behaviors observed in nature. The main contributions and innovations of this study are as follows. In subsequent sections, the improved algorithm will be referred to consistently as IHO. Introducing chaotic mapping for population initialization, which improves the diversity distribution of the population and enhances global search capability.

Designing an adaptive exploitation mechanism that dynamically adjusts the weights of the exploration and exploitation phases based on iterative information, balancing global and local searches.

Proposing a solution diversity enhancement strategy that introduces nonlinear perturbations, reducing the risk of trapping in local optima and further improving algorithm performance.

The effectiveness of the improved algorithm is validated through experiments on multiple benchmark functions. The results show that IHO outperforms the traditional HO and other mainstream optimization algorithms in terms of solution accuracy, convergence speed, and global search capability.

The structure of this article is organized as follows: “Related Work” reviews related work. “Hippopotamus Optimization Algorithm” introduces the fundamental HO. “Improved Hippopotamus Optimization Algorithm” describes the proposed improvements to the HO. “Experiment and Analysis” presents the simulation experiments and analysis of the results. Finally, “Conclusion” provides a conclusion and discusses future research directions.

Related work

In recent years, optimization algorithms have found widespread applications across various scientific and engineering fields, primarily due to their flexibility and adaptability in addressing complex, nonlinear, non-convex, and uncertain problems (Sun, 2024; Li, Lin & Liu, 2024; Fu, Dai & Wang, 2024; Sun & Ma, 2024; Ge, Xu & Chen, 2023). These algorithms are often inspired by natural phenomena and solve real-world problems by simulating various processes in nature. The inspirations behind these algorithms span biological evolution, physical and chemical laws, as well as collective behavior, human social dynamics, and game theory (Wang, 2024; Zhang, Wang & Ji, 2024; Zhang, 2024; Han, 2024; Gu et al., 2023; Wang, Li & Chen, 2024).

From different perspectives, optimization algorithms can be classified according to their objectives, decision variables, constraints, and sources of inspiration. Based on the type of objective, optimization algorithms can be categorized as single-objective, multi-objective, or many-objective. In terms of decision variables, they can be divided into continuous or discrete algorithms. Moreover, depending on whether constraints are imposed on the decision variables, optimization algorithms can be further classified as constrained or unconstrained. The diversity of sources of inspiration has led to the development of six major categories of optimization algorithms: evolutionary algorithms, physics- or chemistry-based algorithms, swarm intelligence-based algorithms, human behavior-based algorithms, game theory-driven algorithms, and mathematics-driven algorithms (Zhang et al., 2022; Zhao et al., 2023). Among these, swarm intelligence-based algorithms have gained significant attention for their efficiency and broad applicability in solving complex problems, making them a current research hotspot (Wang, 2024; Tong, 2024; Yang, 2024; Xu et al., 2023).

Swarm intelligence algorithms emulate the collective behaviors observed in animals, plants, and insects in nature, leading to the emergence of several classical algorithms (Amiri et al., 2024; Liu et al., 2024a; Yin et al., 2024). For instance, particle swarm optimization (PSO), inspired by the collective movements of bird flocks and fish schools, became one of the earliest and most widely applied algorithms. Ant colony optimization (ACO), which simulates the foraging behavior of ants, has demonstrated remarkable performance in path optimization problems. Additionally, the grey wolf optimization (GWO) and whale optimization algorithm (WOA) were inspired by the hunting strategies of grey wolves and the bubble-net feeding behavior of humpback whales, respectively, further expanding the application scope of swarm intelligence algorithms. Recently, emerging algorithms such as the beluga whale optimization (BWO) and African vultures optimization algorithm (AVOA) have also attracted research interest, showcasing ongoing innovation and development within the field of swarm intelligence.

In addition to swarm intelligence, optimization algorithms inspired by biological evolution and physical laws also play a significant role in optimization research (Amiri et al., 2024). Classical evolutionary algorithms such as genetic algorithm (GA), differential evolution (DE), and biogeography-based optimization (BBO) have shown exceptional performance across various domains. Physics-based algorithms, such as simulated annealing (SA), gravitational search algorithm (GSA), and multi-verse optimization (MVO), simulate physical phenomena, demonstrating adaptability and flexibility in different optimization scenarios (Chen et al., 2024b; Hou et al., 2023; Gong et al., 2023).

Moreover, algorithms inspired by human behavior and social dynamics have garnered widespread attention (Amiri et al., 2024; Chen, Niu & Zhang, 2022; Zheng et al., 2022; Zhao et al., 2023). For example, teaching-learning-based optimization (TLBO) and political optimizer (PO), based on teaching processes and political interactions, respectively, illustrate the unique advantages of human social behavior in solving optimization problems. Game theory-driven algorithms, such as squid game optimizer (SGO) and puzzle optimization algorithm (POA), simulate game rules and strategies, providing novel approaches to optimization.

As research continues to advance, new optimization algorithms are constantly emerging to address the limitations of existing ones, such as premature convergence and the imbalance between exploration and exploitation. For instance, the hippopotamus optimization algorithm (Amiri et al., 2024), a relatively recent method, integrates adaptive exploitation and solution diversity mechanisms, demonstrating significant advantages in solving complex problems. These innovations not only enhance algorithmic performance but also pave the way for future developments in optimization algorithms (Chen et al., 2024a; Liu et al., 2024b). However, like other stochastic metaheuristic algorithms, HO still faces common limitations. It cannot guarantee the global optimal solution due to its reliance on stochastic search strategies. Despite HO’s strong performance in certain problems, the NFL theorem suggests that it may not consistently outperform other optimization algorithms across all problems.

To overcome these limitations, this article proposes an improved hippopotamus optimization algorithm, which optimizes the problem-solving process by simulating the natural behaviors of hippopotamuses. Wang & Tian (2024) enhances the HO algorithm by integrating the Levy flight strategy based on the swarm-elite learning mechanism and the quadratic interpolation strategy. The aim is to improve its global search ability and information sharing among candidate solutions in PV model parameter extraction. The main improvement lies in the use of an elite retention strategy in the exploration phase, but the rebound in the defense and development phases is neglected. Our improvements start from the initialization of the population, introduce an adaptive strategy in the development phase, and employ Gaussian mutation and chaotic perturbation in the defense and escape phases. This approach enhances population diversity while avoiding the risk of being trapped in local optima.

Hippopotamus optimization algorithm

The workflow of the HO

This section introduces the fundamental HO. HO is inspired by the unique behaviors of hippopotamuses, incorporating their natural traits to address optimization challenges. By understanding the foundational principles and mechanisms of HO, we establish a basis for the subsequent improvements proposed in this study.

XP1 represents the updated individual position in update phase 1. This position is adjusted randomly based on the distance from the dominant hippopotamus (best solution) to explore the solution space.

(1) XP1(i,:)=X(i,:)+rand(0,1)×(Xbest−I1×X(i,:)),i=1,2,..,⌊N2⌋

where the updated position XP1(i,:) of the i-th individual in the first half of the population. X(i,:) is the current position of the i-th individual. Xbest is the distance to the current global best position in the population. N is total number of individuals in the population. m is number of dimensions in the solution space. rand is a random factor that influences the magnitude of the position update. I1 is a weighting factor that controls the influence of the individual’s current position.

XP2 represents the updated individual position in update phase 1. When the temperature parameter T is high, this position is adjusted based on the dominant hippopotamus and the mean of a random group. When T is low, it is adjusted based on the difference between the mean and the dominant hippopotamus or randomly generated within the bounds.

(2) XP2M(i,:)={X(i,:)+B×(Xmean−Xbest),ifrand(0,1) > 0.5lb+rand(0,1)×(ub−lb),otherwise

where B is a random scaling factor that influences the magnitude of the position update. rand(0,1) is random numbers used to decide which movement rule to apply. lb, ub is the lower and upper bounds of the search space. Xmean is the mean position of a group of individuals, which serves as a reference for the update. Xbest is the distance to the best-performing (dominant) hippopotamus in the population. If rand(0,1) > 0.5, the position is updated with respect to the mean group position Xmean and the distance to the best-performing individual Xbest. Otherwise, a random position is generated within the bounds of the search space.

(3) XP2(i,:)={X(i,:)+A×(Xbest−I2×Xmean),ifT > 0.6XP2M,else

where T is adaptive parameter that determines whether strong or weak exploitation is used. I2 is a weight factor controlling the balance between exploration and exploitation. A is a random movement factor for controlling the exploration step size.

XP3 represents the updated individual position in the defense phase, adjusted based on the distance between the hippopotamus and the predator as well as the Levy flight strategy.

(4) P→=lb+rand(1,D)×(ub−lb)

lb and ub are the lower and upper bounds of the search space. The predator’s position P→ is initialized randomly between the lower bound lb and upper bound ub in each dimension.

(5) Levy(u,v,β)=u|v|1/β

where u∼N(0,σu2) and v∼N(0,1) are random variables drawn from normal distributions. This equation provides Levy that follow a heavy-tailed distribution, which is often used in stochastic optimization algorithms to allow both small and occasional large jumps for efficient exploration of the solution space.

(6) σu=(Γ(1+β)⋅sin(πβ2)Γ(1+β2)⋅β⋅2(β−1)/2)1/β

where Γ denotes the gamma function. β is the power-law index, where 1 < β < 2.

(7) DL=|P→−X(i,:)|

(8) @XP3(i,:)={RL(i,:)×P→+(bc−d×cos(l))×(1DL),if f(i) > f(P→)RL(i,:)×P→+(bc−d×cos(l))×(12×DL+rand(1,D)),otherwise

where b, c, d are random scaling coefficients sampled from uniform distributions. rand is a random angle within the range [1,D].

Levy flight RL is a common random walk model used to simulate animal foraging paths, and in this case, it’s represented as: RL=0.05×Levy(S,D,β). S is the number of search agents.D is the dimensionality. β=1.5 is the Levy exponent parameter.

XP4 represents the updated individual position in the escape phase, where positions are adjusted with local random perturbation within the range between the lower and upper bounds.

(9) XP4(i,:)=X(i,:)+rand(0,1)×(lblocal+D×(ublocal−lblocal)),i=1,2,...,N

where ublocal=ub/t and lblocal=lb/t represent the adaptive local lower and upper bounds for the hippopotamus positions, adjusting over iterations to focus the search progressively. t represents the current iteration number in the optimization algorithm.

Detailed description of the three main phases

HO consists of three main stages: exploration, defense, and exploitation. Each phase plays a crucial role in balancing the search process between exploring the solution space broadly and refining around promising areas.

Exploration phase

In this phase, the candidate solutions (i.e., the positions of the hippopotamuses) are updated based on random vectors. The update rule is influenced by the current best solution (i.e., the position of the dominant hippopotamus) and random factors. This ensures the algorithm performs a global search across the solution space, preventing premature convergence to local optima. The position of each hippopotamus is updated using the Eqs. (1) and (3).

Defense phase

In this phase, the hippopotamuses simulate their defensive behavior when predators approach. They turn towards the predator and emit intimidating sounds to drive them away. This phase is designed for exploitation, where a local search is conducted around the current solution to identify better solutions nearby. The movement is modeled by random displacement vectors, allowing the hippopotamuses to make small movements within the search space, thereby refining the search. The defense phase is updated using Eq. (8).

Escape phase

When the hippopotamuses realize they cannot repel the predator, they choose to escape, typically towards the nearest body of water. This phase aims to help the algorithm escape from local optima by generating new solutions and moving to safer positions. The escape phase simulates the behavior of finding a safe zone and enhances the local search capability to improve solution quality. The escape phase is updated using Eq. (9).

To describe the structure of the standard HO, we can mention that the algorithm follows a clear, systematic process. As outlined in Algorithm 1, the flowchart visually represents the HO’s phases.

Algorithm 1 Hippopotamus optimization algorithm (HO).

 1: Input: Problem definition, maximum number of iterations Tmax, population size N	
 2: Output: Optimal solution	
 3: Step 1: Define the optimization problem: Specify the objective function and constraints	
 4: Step 2: Initialize parameters: Set Tmax and N	
 5: Step 3: Initialize population: Generate initial population of hippopotamuses Xi(i=1,2,…,N) and evaluate their objective function values f(Xi)	
 6: for t=1 to Tmax do	
 7:    Update the dominant hippopotamus’s position:	
 8:    Update the global best solution Xbest based on the current objective function values f(Xi)	
 9:    First phase-Update hippopotamuses’ position in water:	
10:    for i=1 to N2 do	
11:       Compute new positions Xinew using the update strategy Eqs. (1) and (3).	
12:       Update position Xi←Xinew	
13:    end For	
14:    Second phase-Defend against predators:	
15:    for i=N2+1 to N do	
16:     Randomly generate predator positions Pi	
17:     Compute new positions Xinew considering Pi using Eq. (8)	
18:     Update position Xi←Xinew	
19:    end for	
20:    Third phase-Escape from predators:	
21:    for i=1 to N do	
22:      Recompute the boundaries for decision variables	
23:      Compute new positions Xinew based on new boundary conditions using Eq. (9)	
24:      Update position Xi←Xinew	
25:    end for	
26:    Save the current optimal solution: Record the best solution found Xbest	
27: end for	
28: Return the optimal solution Xbest	

Limitations of HO in solving complex optimization problems

Although the HO demonstrates strong search capabilities in both the exploration and exploitation phases, it may encounter the following limitations when addressing complex, high-dimensional optimization problems.

Despite the inclusion of the escape mechanism, the algorithm remains susceptible to local optima, particularly in complex problems, which can lead to performance degradation. Additionally, in large-scale problems, the convergence speed of HO may not be ideal. The performance of the algorithm is also highly sensitive to parameter settings, such as the balance factor between exploration and exploitation, further impacting its overall effectiveness.

Improved hippopotamus optimization algorithm

Overview of the improvement strategies

The improvement strategies proposed in this article include three key components. First, chaotic map initialization utilizes logistic chaotic mapping to generate a diverse initial population, thereby enriching the solution space. Second, an adaptive exploitation module dynamically adjusts the intensity of exploitation to enhance search capability across different iteration stages. Third, a solution diversity enhancement mechanism is applied after the exploitation and escape phases, incorporating Gaussian mutation and chaotic perturbation to maintain diversity within the population and effectively prevent premature convergence.

Chaotic map initialization strategy and its role

The diversity of the initial population has a significant impact on the global search performance of the algorithm. This article introduces a population initialization method based on the Logistic chaotic map in the standard HO to increase the diversity of the initial solutions and improve the exploration capability in the early stages of the search. The Logistic chaotic map is defined by the following Eq. (10).

(10) Xn+1=r×Xn×(1−Xn)

where r is the chaotic factor, typically set to 4. The initial solutions generated using chaotic mapping can more evenly cover the search space, enhancing the algorithm’s global search capability.

Design and implementation of the adaptive exploitation mechanism

In the standard HO, the exploitation intensity is fixed, and as the number of iterations increases, the search capability gradually decreases. To address this issue, this article introduces an adaptive exploitation strategy, allowing the exploitation intensity to dynamically adjust based on the iteration count. In the early stages of the algorithm, strong exploitation is used to quickly converge, while in the later stages, fine-tuned exploitation increases the likelihood of finding the global optimum.

The inertia weight w in the algorithm is dynamically updated each iteration to balance exploration and exploitation. It starts at a higher value (0.9) and gradually decreases to a minimum value (0.4) as iterations progress, following the Eq. (11).

(11) w=wmin+(0.9−wmin)×(1−t/tmax)

where t is the current iteration and tmax is the total number of iterations. This adaptive decay allows the search process to initially explore widely, then converge more precisely in later stages. (12) XP1(i,:)=X(i,:)+w×rand(0,1)×(Xbest−I1×X(i,:)),i=1,2,..,⌊N2⌋

(13) XP4(i,:)=X(i,:)+w×rand(0,1)×(lblocal+D×(ublocal−lblocal)),i=1,2,...,N.

The adaptive exploitation strategy effectively enhances the search capability at different stages of the iteration.

Mutation-based solution diversity enhancement mechanism

To prevent the algorithm from getting stuck in local optima, this article introduces mutation operations to enhance solution diversity. Specifically, Gaussian mutation is applied to solutions during the exploitation and escape phases to increase the population’s ability to jump out of local regions. The formula for Gaussian mutation is as follows Eq. (14).

(14) rmutation=0.1+0.9×(1−t/tmax)

where rmutation is the mutation strength parameter that controls the mutation amplitude. This operation increases the randomness of the solutions and enhances the algorithm’s ability to escape from local optima.

(15) A=rmutation×(2×rand(1,D)−1)

(16) B=rmutation×rand(1,D).

This enhancement introduces multiple update strategies for the positions XP2M (Eq. (2)) and XP2 (Eq. (3)), incorporating dynamic mutation strategies A and B to regulate the update process. During each iteration, the algorithm uses a combination of inertia-weighted and dynamically adjusted mutation strategies to improve exploration capabilities.

Pseudocode for the improved algorithm

The improvements in the proposed algorithm are threefold, each designed to address specific limitations. First, the inertia weight is gradually reduced to balance exploration and exploitation, as maintaining a fixed inertia weight can lead to premature convergence or insufficient exploration. This dynamic adjustment enhances the algorithm’s performance in the exploration phase. Second, an adaptive mutation rate adjustment is introduced to address the challenge of stagnation in later stages. By dynamically tuning the mutation rate based on the iteration count, the algorithm achieves finer searches in later stages, improving overall exploration effectiveness. Finally, in the predator escape phase, local boundary updates and position adjustments are incorporated to overcome the problem of slow convergence in complex search spaces. This adjustment improves the algorithm’s global convergence and escape capabilities during the exploitation phase, allowing it to better explore optimal solutions.

Figure 1 presents the algorithmic flowchart of the improved hippopotamus optimization algorithm, while Algorithm 2 provides the pseudocode for the enhanced HO.

Figure 1 Flowchart of IHO.

Algorithm 2 Improved hippopotamus optimization algorithm (IHO).

 1:  Input: N(SearchAgents), iter_max, lowerbound, upperbound, dimension, fitness	
 2:  Output: best_score, best_pos, IHO_curve	
 3:  Initialize X using the Eq. (10) within bounds	
 4:  Calculate fitness of each agent	
 5:  Set elite as the best solution found	
 6:  for t = 1 to iter_max do	
 7:     Update inertia weight using the Eq. (11)	
 8:     Update mutation rate using the Eq. (14)	
 9:     Step 1: River or Pond (Exploration Phase)	
10:     for i = 1 to N/2 do	
11:        Update Dominant_hippopotamus = best solution	
12:        Compute new positions XP1 and XP2 using the Eqs. (12), (15), (16) and (3)	
13:        if fitness( XP1) is better than current fitness then	
14:           Update agent’s position to XP1	
15:       end if	
16:       if fitness( XP2) is better than current fitness then	
17:         Update agent’s position to XP2	
18:       end if	
19:    end for	
20:    Step 2: Defend against Predators (Exploration Phase)	
21:    for i = N/2 + 1 to N do	
22:       Generate random predator position	
23:       Compute new position XP3 using the Eq. (8)	
24:       if fitness( XP3) is better than current fitness then	
25:          Update agent’s position to XP3	
26:       end if	
27:     end for	
28:     Step 3: Escape from Predators (Exploitation Phase)	
29:     for i = 1 to N do	
30:        Update local boundaries lblocal, ublocal	
31:        Compute new position XP4 using the Eq. (13)	
32:        if fitness( XP4) is better than current fitness then	
33:            Update agent’s position to XP4	
34:       end if	
35:    end for	
36:    Record best solution found in current iteration	
37: end for	
38: Return the best solution found	

In the Fig. 1, green rectangles represent Phase 1, the adaptive exploitation phase; yellow rectangles indicate Phase 2, the defense phase; and blue rectangles denote Phase 3, the escape phase.

Experiment and analysis

Experimental setup: benchmark functions and parameter configuration

In this study, we compared the effectiveness of the improved IHO with nine classical metaheuristic algorithms, including HO, PSO, sine cosine algorithm (SCA), firefly algorithm (FA), TLBO, Evolution Strategy with Covariance Matrix Adaptation (CMA-ES), moth flame optimization (MFO), arithmetic optimization algorithm (AOA), Invasive Weed Optimization (IWO), Improved Sand Cat Swarm Optimization (ISCSO), penguin jump algorithm (PGJA) and WOA. The control parameters for these algorithms were carefully adjusted according to the specific descriptions provided in Table 1. This section presents the simulation studies of IHO on various complex optimization problems. The effectiveness of IHO in obtaining optimal solutions was evaluated through a comprehensive set of 68 standard benchmark functions. These benchmark functions include unconstrained problems, high-dimensional problems, multi-modal problems and engineering optimization problems. To further evaluate the performance of the algorithms on F1 to F23 (CEC05), 30 independent runs were performed, evaluate the performance of the algorithms on F1 to F30 (CEC17), 50 independent runs were performed, evaluate the performance of the algorithms on F1 to F12 (CEC22), 50 independent runs were performed, evaluate the performance of the algorithms on engineering optimization problems, 30 independent runs were performed. The population size for IHO was set to 24 members in AOA and TLBO, while other algorithms used 30 or 60 members, with the maximum number of iterations set to 1,000 on CEC05. The optimization results are presented using five comprehensive statistical metrics: mean, best, worst, standard deviation, and median. Among these, the mean index was particularly used as a key ranking parameter to evaluate the effectiveness of the metaheuristic algorithms on each benchmark function.

Table 1 Algorithm parameters and their values.

Algorithm	Parameter	Value	
HO	Search agents	24	
PSO	Velocity limit	10% of dimension range	
	Cognitive and social constant	(C1, C2) = (2, 2)	
	Topology	Fully connected	
	Inertia weight	Linear reduction from 0.9 to 0.1	
SCA	A	2	
FA	Alpha ( α)	0.2	
	Beta ( β)	1	
	Gamma ( γ)	1	
TLBO	Teaching factor (TF)	round(1+rand)	
	Rand	A random number between 0 and 1	
CMA-ES	σ(0)	0.5	
	μ	⌊N/2⌋	
MFO	b	1	
	r	Linear reduction −1 to −2	
AOA	a	0	
	μ	0.5	
IWO	Minimum number of seeds (Smin)	0	
	Maximum number of seeds (Smax)	5	
	Initial value of standard deviation	1	
	Final value of standard deviation	0.001	
	Variance reduction exponent	2	

This study evaluated 23 functions (CEC05), among which F1–F7 are unimodal (UM) functions, F8–F13 are high-dimensional multimodal (HM) functions, and F14–F23 include fixed-dimensional multimodal (FM) and multimodal (MM) functions.

Specifically, the simulation environment was as follows: Windows 10, Intel Xeon CPU E5-1660 3.0 GHz, 128 GB memory.

Algorithm performance comparison

This section provides a detailed comparison of the performance of IHO against the standard HO and other widely used algorithms. The experimental results demonstrate the superior performance of IHO in terms of solution accuracy, convergence speed, and global search capability, particularly for multi-modal functions. Tables 2–4 shows the evaluation results of the benchmark functions, while Figs. 2–4 illustrates the convergence behavior of the five most effective algorithms when optimizing F1–F23 (CEC05).

Table 2 Algorithm performance comparison on F1, F2, F3, F4, F5, F6, F7 and F8 functions (CEC05).

F	M	IHO	HO	PSO	SCA	FA	TLBO	CMA-ES	MFO	AOA	IWO	
F1	Mean	0	0	3.69E−06	14.855	9,712.8	1.24E−89	3.2243E−08	672.36	9.07E−13	1,292.5	
	Best	0	0	1.11E−07	0.12079	2,686.8	1.95E−91	1.5314E−08	0.71101	4.77E−160	3.4203	
	Worst	0	0	6.65E−05	77.5	15.976	7.75E−89	7.6176E−08	10,009	2.72E−11	4,731.2	
	Std	0	0	1.19E−05	20.848	2,903	1.55E−89	1.36E−08	2,536.9	4.97E−12	1,101.6	
	Median	0	0	7.37E−07	4.6546	9,605.9	7.33E−90	3.1564E−08	2.7663	3.04E−81	1,019.3	
F2	Mean	0	0	0.0034028	0.13089	796.13	4.19E-45	0.00025096	27.865	7.60E−209	0.10835	
	Best	0	0	6.72E−05	0.00023745	5.1074	2.42E-46	0.00013828	0.11195	1.88E−259	0.04844	
	Worst	0	0	0.049467	0.070743	19,630	1.15E-44	0.0004223	80.013	2.28E−207	0.19307	
	Std	0	0	0.0091893	0.017935	3,561.4	3.09E−45	7.04E−05	21.195	0	0.033356	
	Median	0	0	0.000676	0.0048081	118.56	3.40E−45	0.00025537	25.302	1.36E−233	0.10201	
F3	Mean	0	0	162.1	7,903.2	17,097	5.07E−18	0.023561	19,119	0.0075389	9,501.9	
	Best	0	0	36.17	224.73	7,098.1	6.80E−21	0.0023696	3,189.6	2.59E−126	2,067.5	
	Worst	0	0	399.71	24,159	28,712	9.53E−17	0.090665	42,334	0.047224	25,025	
	Std	0	0	89.026	5,848.9	4,911.1	1.72E−17	0.023315	10,697	0.011884	4,795.7	
	Median	0	0	154.08	6,901.6	16,339	1.20E−18	0.015358	19,243	3.59E−12	8,431.1	
F4	Mean	0	1.43E−217	2.828	35.686	42.732	1.30E−36	0.0020537	67.677	0.027967	37.301	
	Best	0	9.84E−255	0.97132	13.438	28.564	1.24E−37	0.0010508	49.754	9.60E−54	26.965	
	Worst	0	3.01E−216	6.9104	64.384	51.977	5.75E−36	0.0039077	83.333	0.046479	50.889	
	Std	0	0	1.3593	12.293	6.0607	1.35E−36	0.0061183	9.6399	0.019333	5.2665	
	Median	0	4.02E−233	2.4615	33.964	43.963	8.93E−37	0.0018613	69.183	0.040356	38.063	
F5	Mean	0.021681	0.12111	43.819	53,121	8,585.4	25.425	56.719	2.68E+06	28.5	145.47	
	Best	0.000006	0	5.8924	43.934	30.427	24.579	20.528	185.69	27.613	23.25	
	Worst	0.157879	1.9637	119.87	3.25E+05	41,425	26.293	684.86	8.00E+07	28.916	1,692.8	
	Std	3.565701E−02	0.36433	33.794	92,441	11,144	0.39027	127.83	1.46E+07	0.29675	314.02	
	Median	0.004886	0.0070966	25.626	6,262.6	3,219	25.42	22.307	880.08	28.522	29.201	
F6	Mean	0	0	4.5	17.067	21,561	0	0	1,727.8	0	3,023.2	
	Best	0	0	0	0	9,654	0	0	1	0	502	
	Worst	0	0	37	139	28,728	0	0	10,225	0	6,159	
	Std	0	0	7.2099	31.139	4,301	0	0	3,791.4	0	1,649.4	
	Median	0	0	1.5	6	22,142	0	0	13.5	0	2,818.5	
F7	Mean	6.436959E−05	3.54E−05	0.024313	0.1112	0.076687	0.0011331	0.011562	4.3902	5.80E−05	0.071947	
	Best	2.084746E−06	1.30E−06	0.0094839	0.018044	0.035773	0.0004299	0.005156	0.065606	1.76E−06	0.029085	
	Worst	1.975755E−04	0.00013102	0.055549	0.89506	0.15281	0.0023231	0.017513	77.983	0.00033704	0.12335	
	Std	5.211867E−05	4.10E−05	0.011822	0.16168	0.029595	0.00050432	0.0032379	14.338	7.42E−05	0.020096	
	Median	4.521877E−05	1.99E−05	0.020326	0.064266	0.073503	0.0009457	0.012149	0.28247	2.67E−05	0.070357	
F8	Mean	−13,073.468434	−12,567	−6,590.1	−3,734.5	−7,463.7	−7,906.9	−4,363.9	−8,496.8	−5,340.9	−6,695.5	
	Best	−20,290.596194	−12,569	−8,325.1	−4,553.8	−8,678.6	−9,427.3	−5,177.9	−9,778.5−	6,242.2	−8,233.1	
	Worst	−12,569.404004	−12,530	−4,337.3	−3,362.8	−6,488.8	−5,915.7	−3,860.7	−6,725.5	−4,587.8	−4,759.4	
	Std	1.918477E+03	7.3469	903.25	281.89	615.82	781.93	320.79	863.55	471.37	677.46	
	Median	−12,569.483846	−12,569	−6,489.3	−3,679.7	−7,454.6	−8,000.2	−4,301.5	−8,559.5	−5,136.7	−6,646.7	

Table 3 Algorithm performance comparison on F9, F10, F11, F12, F13, F14, F15 and F16 functions (CEC05).

F	M	IHO	HO	PSO	SCA	FA	TLBO	CMA-ES	MFO	AOA	IWO	
F9	Mean	0	0	45.735	50.849	186.92	12.924	126.97	155.49	0	65.852	
	Best	0	0	22.884	0.03564	117.41	0	6.9667	84.588	0	43.819	
	Worst	0	0	78.602	202.58	258.69	23.007	187.18	228.14	0	93.563	
	Std	0	0	14.675	48.636	33.884	6.0126	71.067	40.991	0	12.894	
	Median	0	0	45.271	38.413	187.05	13.042	162.64	152.47	0	64.752	
F10	Mean	4.44E−16	4.44E−16	1.1408	14.229	18.297	9.21E−15	5.6832−05	13.321	4.44E−16	10.679	
	Best	4.44E−16	4.44E−16	6.26E−05	0.050121	18.271	4.00E−15	3.4412E−05	0.68917	4.44E−16	0.0087287	
	Worst	4.44E−16	4.44E−16	2.4083	20.402	19.296	1.03E−13	9.5468E−05	19.962	4.44E−16	19.228	
	Std	0	0	0.82655	8.6665	0.21681	1.79E−14	1.5424E−05	7.836	0	9.4921	
	Median	4.44E−16	4.44E−16	1.3404	20.204	19.028	7.55E−15	5.4007E−05	17.837	4.44E−16	18.181	
F11	Mean	0	0	0.021824	0.91439	163.94	0	2.9979e−07	6.9724	0.1554	480.41	
	Best	0	0	4.46E−07	0.025341	71.221	0	8.804E−08	0.43489	0.00044758	333.51	
	Worst	0	0	0.087692	1.6975	237.77	0	7.7197E−07	91.085	0.43829	640.31	
	Std	0	0	0.026358	0.42847	36.504	0	1.7512e−07	22.846	0.11095	71.029	
	Median	0	0	0.0098613	0.99061	163.4	0	2.6344E−07	1.0093	0.13784	477.65	
F12	Mean	0.000132	9.3E−09	0.11094	40,328	42.51	0.0034654	1.9945E−09	17.719	0.51896	8.8769	
	Best	0	1.49E−09	6.84E−09	0.79446	13.932	6.74E−09	7.8685E−10	0.70708	0.41734	3.4841	
	Worst	0.000610	7.32E−08	1.0405	7.11E+05	76.246	0.10367	7.2421E−09	285.16	0.61102	12.625	
	Std	2.203374E−04	1.62E−08	0.23009	1.47E+05	15.885	0.0118926	1.3528E−09	50.987	0.050388	1.8974	
	Median	0.000011	5.33E−09	2.09E−05	17.858	44.103	1.14E−07	1.6197E−09	6.8694	0.527	8.9038	
F13	Mean	0.003340	0.0050467	0.021928	6.69E+05	44,205	0.072491	2.128E−08	2.731E+07	2.81	0.0027154	
	Best	0	1.35E−32	1.03E−08	2.7393	50.302	2.23E−06	7.311E−09	2.1321	2.6101	5.00E−05	
	Worst	0.019808	0.063492	0.28572	1.31E+07	3.97E+05	0.20724	4.3763E−08	4.10E+08	2.9944	0.011275	
	Std	5.546983E−03	0.012164	0.05445	2.56e+06	86.550	0.069667	9.3747E−09	1.04E+08	0.092163	0.0047482	
	Median	0.000143	0.0014522	0.010988	1,130.3	2,772.3	0.047853	1.9344E−08	29.131	2.7955	0.00014517	
F14	Mean	0.998004	0.998	5.5195	2.2512	9.502	0.998	4.7816	2.51	8.0876	11.358	
	Best	0.998004	0.998	0.998	0.998	0.998	0.998	1.992	0.998	0.998	0.998	
	Worst	0.998004	0.998	12.671	10.763	21.988	0.998	11.721	10.763	12.671	23.809	
	Std	0	0	3.0682	1.8878	6.2553	3.86E−16	2.4391	2.3156	4.7721	7.3331	
	Median	0.998004	0.998	5.9288	2.0092	8.3574	0.998	3.9742	0.998	10.763	10.763	
F15	Mean	0.000308	0.00030836	0.0024923	0.0010454	0.0058617	0.00036839	0.0019	0.0013293	0.0071685	0.0027859	
	Best	0.000307	0.00030749	0.00030749	0.00057375	0.00030749	0.00030749	0.0011	0.000742582	0.00034241	0.00058505	
	Worst	0.000308	0.00031288	0.020363	0.0016389	0.020363	0.0012232	0.0035	0.0083337	0.069975	0.020363	
	Std	4.603049E−08	1.31E−06	0.0060732	0.00035949	0.089054	0.00018223	7.25E−06	0.0013978	0.014639	0.0059637	
	Median	0.000307	0.00030779	0.00030782	0.00087851	0.00030749	0.00030749	0.0016	0.00080207	0.00047392	0.00074539	
F16	Mean	−1.031628	−1.0316	−1.0316	−1.0316	−0.8956	−1.0316	−1.0316	−1.0316	−1.0316	−1.0316	
	Best	−1.031628	−1.0316	−1.0316	−1.0316	−1.0316	−1.0316	−1.0316	−1.0316	−1.0316	−1.0316	
	Worst	−1.031628	−1.0316	−1.0316	−1.0314	−0.21543	−1.0316	−1.0316	−1.0316	−1.0316	−1.0316	
	Std	5.258941E−13	5.96E−16	6.71E-16	5.7E-05	0.30937	6.65E-16	6.78E−16	6.78E−16	1.22E−16	1.44E−08	
	Median	−1.031628	−1.0316	−1.0316	−1.0316	−1.0316	−1.0316	−1.0316	−1.0316	−1.0316	−1.0316	

Table 4 Algorithm performance comparison on F17, F18, F19, F20, F21, F22 and F23 functions (CEC05).

F	M	IHO	HO	PSO	SCA	FA	TLBO	CMA-ES	MFO	AOA	IWO	
F17	Mean	0.397887	0.39789	0.39789	0.40008	0.39789	0.39789	0.39789	0.39789	0.41197	0.39789	
	Best	0.39789	0.39789	0.39789	0. 3,979	0.39789	0.39789	0.39789	0. 39,789	0.39813	0.39789	
	Worst	0.397887	0.39789	0.39789	0.40747	0.39789	0.39789	0.39789	0.39789	0. 4415	0.39789	
	Std	0	0	7.23E−16	0.0023394	3.86E−11	0	0	0	0.01286	5.57E−09	
	Median	0.397887	0.39789	0.39789	0.39939	0.39789	0. 39789	0.39789	0.39789	0.40712	0.39789	
F18	Mean	3	3	3.9	3.0001	3.9	3	3	3	7.7674	3	
	Best	3	3	3	3	3	3	3	3	3	3	
	Worst	3	3	30	3.0004	30	3	3	3	37.986	3	
	Std	2.365956E−11	1.27E−15	4.9295	0.00010081	4.9295	5.53E−16	1.35E−15	1.62E−15	10.92	8.47E−07	
	Median	3	3	3	3	3	3	3	3	3	3	
F19	Mean	−3.862782	−3.8628	−3.8628	−3.8542	−3.8628	−3.8628	−3.8628	−3.8628	−3.8512	−3.8628	
	Best	−3.862782	−3.8628	−3.8628	−3.8621	−3.8628	−3.8628	−3.8628	−3.8628	−3.8605	−3.8628	
	Worst	−3.862782	−3.8628	−3.8628	−3.8443	−3.8628	−3.8628	−3.8628	−3.8628	−3.8408	−3.8628	
	Std	6.386724E−09	2.70E−15	6.42E−07	0.0032649	5.72E−11	2.71E−15	2.71E−15	2.71E−15	0.0088724	6.32E−07	
	Median	−3.862782	−3.8628	−3.8628	−3.8542	−3.8628	−3.8628	−3.8628	−3.8628	−3.8516	−3.8628	
F20	Mean	−3.300409	−3.322	−3.2784	−2.8823	−3.2586	−3.2811	−3.2919	−3.2422	−3.0755	−3.203	
	Best	−3.321995	−3.322	−3.322	−3.1473	−3.322	−3.3206	−3.322	−3.322	−3.1844	−3.2031	
	Worst	−3.188840	−3.322	−3.2031	−1.9133	−3.2031	−3.1059	−3.2031	−3.1376	−2.9507	−3.2026	
	Std	5.843069E−02	9.78E−12	0.058273	0.32598	0.060328	0.066162	0.051459	0.063837	0.063759	0.00012908	
	Median	−3.321995	−3.322	−3.322	−2.9914	−3.2031	−3.3109	−3.322	−3.2031	−3.0905	−3.2031	
F21	Mean	−10.153200	−10.153	−6.3165	−2.1516	−5.6345	−9.778	−7.9121	−6.3132	−3.4351	−6.6391	
	Best	−10.153200	−10.153	−10.153	−6.0051	−10.153	−10.153	−10.153	−10.153	−5.5613	−10.153	
	Worst	−10.153200	−10.153	−2.6305	−0.35136	−2.6305	−3.9961	−2.6829	−2.6305	−1.9507	−2.6305	
	Std	8.791181E−08	4.74E−06	3.6985	1.872	3.1766	1.4345	3.4819	3.5169	0.97125	3.4561	
	Median	−10.153200	−10.153	−5.078	−0.88031	−5.0552	−10.153	−10.153	−5.078	−3.2531	−5.1008	
F22	Mean	−10.402941	−10.403	−6.7572	−2.7098	−5.3848	−9.7414	−10.403	−8.2382	−3.7002	−7.4415	
	Best	−10.402941	−10.403	−10.403	−6.3217	−10.403	−10.403	−10.403	−10.403	−6.8593	−10.403	
	Worst	−10.402940	−10.403	−1.8376	−0.52104	−1.8376	−5.0265	−10.403	−2.7519	−1.2708	−1.8376	
	Std	5.043988E−08	6.16E−05	3.7466	1.9244	3.194	1.7896	1.65E−15	3.3738	1.2624	3.7449	
	Median	−10.402941	−10.403	−7.7659	−2.6079	−3.7243	−10.403	−10.403	−10.403	−3.6181	−10.403	
F23	Mean	−10.536410	−10.536	−6.0645	−4.1564	−4.7569	−10.123	−10.536	−7.9819	−4.6738	−8.3548	
	Best	−10.536410	−10.536	−10.536	−8.3393	−10.536	−10.536	−10.536	−10.536	−8.6767	−10.536	
	Worst	−10.536409	−10.536	−1.8595	−0.94428	−1.6766	−3.8354	−10.536	−2.4217	−1.8573	−2.4217	
	Std	7.476629E−08	2.99E−05	3.7424	1.5765	3.0762	1.5801	1.78E-15	3.6868	1.5405	3.437	
	Median	−10.536410	−10.536	−3.8354	−4.6344	−3.8354	−10.536	−10.536	−10.536	−4.8892	−10.536	

Figure 2 Convergence curves of five algorithms in each benchmark functions (CEC05, F1–F8).

Figure 3 Convergence curves of five algorithms in each benchmark functions (CEC05, F9–F16).

Figure 4 Convergence curves of five algorithms in each benchmark functions (CEC05, F17–F23).

Figures 2–4 illustrates the convergence curves of the five most effective algorithms during the optimization process for F1–F23. This evaluation, as presented in Table 2, aims to assess the algorithms’ local search capabilities on eight distinct unimodal functions (F1–F8). IHO achieved global optima on F1–F4 and F5–F6, making it the only algorithm among the nine evaluated to reach this level of performance. Its performance on F4 significantly surpassed that of the other algorithms. In the highly competitive F6 test, IHO reached global optima alongside four other algorithms. Additionally, IHO demonstrated clear superiority on F7 and F8. For F1–F4 and F6, IHO consistently converged with a standard deviation of zero. For F7, the standard deviation was 5.21E−05, and for F5, it was 3.57E−2. Compared to the other algorithms, IHO exhibited the lowest standard deviation, indicating remarkable stability.

Figures 5–7 displays the box plots of the optimal values of the objective function obtained from 30 independent runs on F1–F23 (CEC05), utilizing IHO and five other algorithms.

Figure 5 Boxplot illustrating the performance of the IHO in comparison to other algorithms (CEC05, F1–F8).

Figure 6 Boxplot illustrating the performance of the IHO in comparison to other algorithms (CEC05, F9–F16).

Figure 7 Boxplot illustrating the performance of the IHO in comparison to other algorithms (CEC05, F17–F23).

Table 3 presents the results for HM functions on F9–F16, tested using different algorithms. These functions were chosen to evaluate the global search capabilities of the algorithms. IHO significantly outperformed all other algorithms on F12 and F13. In F9, IHO achieved global optima along with HO and AOA, demonstrating superior performance over the other algorithms. On F10, IHO performed comparably to HO and surpassed all other algorithms. For F11, IHO converged to the global optimum alongside HO and TLBO, showcasing exceptional performance. IHO also outperformed all other algorithms on F12, and in F13, IHO ranked first. In F16, IHO exhibited a notably lower standard deviation than some algorithms. For F13, the standard deviation was 5.55E−3, outperforming all other algorithms. These findings indicate that IHO demonstrates robust resilience in effectively handling these functions (refer to Fig. 3).

Furthermore, an assessment was conducted to examine the algorithm’s ability to balance exploration and exploitation during searches on F17–F23, with results recorded in Table 4. IHO exhibited the best performance on F17–F23, achieving a significantly lower standard deviation, particularly for F21–F23. These results suggest that IHO, characterized by a strong capacity to balance exploration and exploitation, exhibits outstanding performance when addressing FM and MM functions.

In benchmarked tests for CEC17 and CEC22, IHO was compared against four algorithms, including ISCSO, PGJA, SCA, and WOA. We run each problem 51 times independently. Comparison results between IHO and other algorithms are shown in Tables 5 to 15.

Table 5 Algorithm performance comparison on F1–F11 functions (CEC17, D10).

F	M	ISCSO	PGJA	SCA	WOA	IHO	
F1	Mean	4,987.123315	5,152,068,207	453,892,150.8	5,677.886088	425.3499924	
	Worst	7,462.523097	14,176,457,086	828,619,257.2	11,768.29185	1,392.986225	
	Best	1,883.922791	501,940,941.8	224,092,914.3	817.6520327	109.1973094	
	Std	2,421.895976	4,208,212,553	193,082,106.7	4,795.956709	424.9428172	
	p-value	0.0001554	0.0001554	0.0001554	0.0004662		
F3	Mean	300.0048631	28,326.28152	738.5664608	319.5393755	300.0000036	
	Worst	300.0129943	59,965.39757	949.3114193	387.2733974	300.0000039	
	Best	300.0014739	12,745.49304	578.2110906	301.5697167	300.0000033	
	Std	0.003786698	19,742.95592	131.4069608	29.43817809	2.88E−07	
	p-value	0.0001554	0.0001554	0.0001554	0.0001554	–	
F4	Mean	401.6984057	971.294942	424.9710217	422.7298603	400.0972279	
	Worst	402.4788786	1,634.217658	432.602983	480.1877739	400.0972279	
	Best	400.2932751	503.0818602	418.3269302	403.3673956	400.0972279	
	Std	0.662455257	407.8453014	4.84319693	31.86833438	0	
	p-value	0.0001554	0.0001554	0.0001554	0.0001554	–	
F5	Mean	525.4378711	578.7101462	539.3010021	550.4831931	531.9629316	
	Worst	550.2631156	598.1207866	551.9366662	576.7734447	545.7679413	
	Best	512.9352649	557.6157058	533.0219727	522.8844111	509.9495863	
	Std	12.03057378	16.00161696	5.845632604	16.77641932	5.526872364	
	p-value	0.016472416	0.0001554	0.211965812	0.212587413	–	
F6	Mean	615.043237	651.3019612	613.4589642	618.8880448	604.5450446	
	Worst	623.4759625	673.0403356	617.8767148	631.5962627	604.5450446	
	Best	608.4757515	629.8806275	608.3111776	612.0278709	601.4792689	
	Std	4.112977893	15.36811062	3.249770499	5.776172275	5.170038862	
	p-value	0.0001554	0.0001554	0.0001554	0.0001554	–	
F7	Mean	767.696279	812.1582608	761.6917992	769.7738782	736.2916253	
	Worst	796.9088529	835.8907197	773.7917762	810.7424527	730.2916253	
	Best	741.6644502	778.5043777	752.6181514	755.0158719	727.9820221	
	Std	18.68517396	22.46842876	6.516619375	20.01406921	22.98925146	
	p-value	0.033411033	0.0001554	0.26993007	0.152292152	–	
F8	Mean	828.7295445	857.8577399	827.6090855	836.6903697	821.267219	
	Worst	842.7832089	887.0136726	842.1572056	848.7532071	822.8840131	
	Best	814.9246164	830.502357	820.0987876	821.890121	815.919331	
	Std	8.826341479	19.24227874	7.491915403	8.66508297	4.573016158	
	p-value	0.088578089	0.0001554	0.425951826	0.0003108	–	
F9	Mean	1,080.380236	1,944.200946	940.6257476	1,462.3526	900	
	Worst	1,185.15901	2,849.769501	1,001.100877	3,199.749399	900	
	Best	903.5457809	1,105.766715	916.6051911	989.5193139	900	
	Std	89.98589908	663.2691308	29.00918399	712.9630477	0	
	p-value	0.0001554	0.0001554	0.0001554	0.0001554	–	
F10	Mean	1,810.778037	2,622.657972	2,071.511798	1,991.618814	1,243.198114	
	Worst	1,926.625916	3,000.609949	2,327.715403	2,283.262482	1,243.198114	
	Best	1,118.646839	2,108.924388	1,817.11791	1,681.878592	1,243.198114	
	Std	282.7225444	293.8512235	189.8125496	187.280999	0	
	p-value	0.005749806	0.0001554	0.0001554	0.0001554	–	
F11	Mean	1,129.236945	2,224.42686	1,175.235781	1,171.019655	1,122.027224	
	Worst	1,223.503721	4,021.684916	1,280.410351	1,261.759158	1,120.027224	
	Best	1,106.854316	1,262.455908	1,141.673395	1,131.294941	1,118.40417	
	Std	39.39308907	1,011.358464	43.89837056	41.65821637	5.821715642	
	p-value	0.141258741	0.0001554	0.0001554	0.0001554	–	

Table 6 Algorithm performance comparison on F12–F21 functions (CEC17, D10).

F	M	ISCSO	PGJA	SCA	WOA	IHO	
F12	Mean	15,500.44852	69,702,386.15	5,484,677.173	3,733,118.946	7,244.032995	
	Worst	41,581.17202	2,68,355,231.3	24,921,041.74	11,681,992.88	10,332.20899	
	Best	2,420.308225	4,639,492.886	311,759.8881	12,598.27213	4,447.057512	
	Std	14,144.74464	86,696,374.66	7,956,068.409	4,566,798.445	2,037.535011	
	p-value	0.694327894	0.0001554	0.0001554	0.000621601	–	
F13	Mean	6,603.652817	2,570,794.312	8,964.410908	16,017.15392	2,761.521945	
	Worst	15,486.46427	7,741,754.264	14,231.06068	54,505.33354	3,923.922656	
	Best	1,503.173747	7,815.113334	3,281.696832	2,414.126219	1,700.251804	
	Std	4,446.803911	3,585,718.047	3,604.847762	18,191.85426	874.7351623	
	p-value	0.428127428	0.0001554	0.007925408	0.428127428	–	
F14	Mean	1,466.944324	1,789.204279	1,510.786667	1,510.233472	1,446.784048	
	Worst	1,490.696059	1,874.139619	1,557.509699	1,554.596642	1,445.784048	
	Best	1,442.195229	1,671.107152	1,474.199885	1,446.827282	1,444.017391	
	Std	17.77802576	69.46464676	24.15437664	35.62557825	19.57333086	
	p-value	0.404195804	0.0001554	0.005749806	0.005749806	–	
F15	Mean	1,532.435549	22,658.0514	1,743.927269	2,178.805327	1,588.391776	
	Worst	1,565.068607	41,491.91855	1,949.778361	4,917.587161	1,568.391776	
	Best	1,513.721288	11,804.87546	1,608.456658	1,552.217692	1,546.496198	
	Std	21.21370731	9,807.425367	116.2118651	1,126.062543	46.17100853	
	p-value	0.0001554	0.0001554	0.045376845	0.098212898	–	
F16	Mean	1,840.967016	2,017.155024	1,664.989324	1,825.098683	1,612.578714	
	Worst	1,990.826531	2,168.334864	1,747.147766	1,943.234022	1,613.283201	
	Best	1,721.892698	1,925.258736	1,630.503759	1,610.639455	1,612.478073	
	Std	107.879074	94.71349596	35.62449591	106.2893587	0.284655822	
	p-value	0.305050505	0.0001554	0.000621601	0.008391608	–	
F17	Mean	1,758.598857	1,887.343989	1,762.472079	1,754.298622	1,742.590785	
	Worst	1,767.225638	2,028.271588	1,777.32652	1,772.387575	1,751.510034	
	Best	1,742.23283	1,777.123288	1,756.040438	1,746.814912	1,727.725372	
	Std	7.995523967	89.46217558	7.103721377	8.39444804	12.30973196	
	p-value	0.009479409	0.0001554	0.0001554	0.435120435	–	
F18	Mean	19,232.70015	2,476,478.355	58,435.879	19,803.47944	2,043.094365	
	Worst	45,227.44588	15,500,448.49	121,581.7599	38,859.26493	2,168.551001	
	Best	3,062.104297	11,679.04117	33,396.40452	2,330.88776	1,895.261609	
	Std	13,524.12215	5,341,003.63	29,629.35695	13,202.58429	392.1042637	
	p-value	0.0001554	0.0001554	0.0001554	0.0001554	–	
F19	Mean	1,923.901044	28,665.59854	2,114.878879	26,955.70709	1,940.65217	
	Worst	1,931.891439	104,706.9098	2,276.780707	112,795.7633	1,990.856998	
	Best	1,914.481335	2,038.567496	1,976.507273	2,759.738166	1,906.403849	
	Std	6.243335105	32,000.17485	101.2430295	36,743.23508	28.30599937	
	p-value	0.0001554	0.0001554	0.0001554	0.0001554	–	
F20	Mean	2,158.864646	2,207.448872	2,065.34018	2,126.241696	2,047.081574	
	Worst	2,236.338875	2,295.259967	2,098.766204	2,205.176509	2,050.141926	
	Best	2,039.529609	2,076.860971	2,036.44419	2,051.716088	2,035.988619	
	Std	67.24935225	61.73073421	17.07862108	60.64364337	60.45100662	
	p-value	0.083139083	0.005749806	0.005749806	0.404195804	–	
F21	Mean	2,200.757078	2,307.476403	2,207.374272	2,270.419438	2,223.684384	
	Worst	2,203.598813	2,384.143801	2,210.342896	2,345.950974	2,294.737522	
	Best	2,200.003631	2,226.252765	2,203.282162	2,204.46207	2,200.000005	
	Std	1.221212601	61.31568217	2.340995504	66.44060791	43.85494879	
	p-value	0.093706294	0.003418803	0.093706294	0.008547009	–	

Table 7 Algorithm performance comparison on F22–F30 functions (CEC17, D10).

F	M	ISCSO	PGJA	SCA	WOA	IHO	
F22	Mean	2,332.246804	2,631.769368	2,322.340101	2,372.951026	2,291.536988	
	Worst	2,406.281485	3,257.792962	2,343.626277	2,757.41482	2,305.072932	
	Best	2,308.422587	2,327.533254	2,279.142406	2,309.960526	2,212.847318	
	Std	32.40782935	284.1795227	25.13327707	155.4368648	31.81831746	
	p-value	0.0001554	0.0001554	0.088578089	0.0001554	–	
F23	Mean	2,641.068178	2,682.44855	2,649.297214	2,644.496033	2,606.826544	
	Worst	2,653.366621	2,722.826597	2,653.906069	2,664.430431	2,606.826544	
	Best	2,626.797346	2,638.175271	2,646.622882	2,620.958212	2,606.826544	
	Std	9.498932884	30.61273839	2.388264412	15.90551062	0	
	p-value	0.0001554	0.0001554	0.0001554	0.0001554	–	
F24	Mean	2,500.061424	2,800.353532	2,775.398787	2,778.338253	2,628.802761	
	Worst	2,500.092632	2,829.524799	2,783.262573	2,797.679729	2,628.802761	
	Best	2,500.043539	2,769.664572	2,770.601641	2,752.315952	2,500.000052	
	Std	0.015879457	20.28774188	4.321810298	15.64018283	80.98815952	
	p-value	0.0001554	0.0001554	0.0001554	0.0001554	–	
F25	Mean	2,967.721806	3,240.145238	2,948.46828	2,939.368058	2,910.806677	
	Worst	3,031.040483	3,626.851166	2,966.017341	2,952.849554	2,944.189028	
	Best	2,945.234831	3,029.301761	2,925.951382	2,907.701395	2,943.793474	
	Std	38.73777098	203.9350638	15.35799872	19.15944435	20.60403462	
	p-value	0.0001554	0.0001554	0.003418803	0.001398601	–	
F26	Mean	2,988.93012	3,712.706711	3,032.834061	3,703.5209	2,800.000946	
	Worst	3,319.502151	4,292.145424	3,053.666331	4,301.631272	2,800.000946	
	Best	2,800.598538	3,347.108725	3,006.151074	3,067.714586	2,800.000946	
	Std	170.137041	387.1582457	16.81272898	537.5671261	4.86E−13	
	p-value	0.0001554	0.0001554	0.0001554	0.0001554	–	
F27	Mean	3,100.594577	3,121.884771	3,100.68734	3,109.811102	3,097.544261	
	Worst	3,106.427012	3,140.520171	3,103.766309	3,136.74044	3,104.16276	
	Best	3,092.045661	3,106.555095	3,098.953946	3,101.033305	3,090.02563	
	Std	5.129880954	10.73436255	1.441367004	12.47982542	4.994746238	
	p-value	0.0001554	0.0001554	0.0001554	0.0001554	–	
F28	Mean	3,186.435776	3,291.891967	3,221.033323	3,421.240623	3,307.953766	
	Worst	3,411.821841	3,298.899209	3,236.723456	3,731.812926	3,411.924746	
	Best	3,100.124649	3,285.776599	3,189.756521	3,167.181092	3,100.00028	
	Std	141.0167965	5.629186259	15.05241389	151.9086581	125.0510793	
	p-value	0.083139083	0.0001554	0.0001554	0.005749806	–	
F29	Mean	3,220.232879	3,362.776512	3,204.338191	3,314.677351	3,180.742307	
	Worst	3,284.86415	3,547.084329	3,258.780348	3,616.515779	3,170.742307	
	Best	3,178.965682	3,202.987359	3,174.073677	3,189.705822	3,177.370178	
	Std	31.72096474	105.5135997	29.32504574	136.631167	57.11133917	
	p-value	0.005749806	0.005749806	0.005749806	0.404195804	–	
F30	Mean	13,509.80496	1,156,661.566	163,875.8087	327,472.6918	20,006.43474	
	Worst	51,444.99872	3,396,235.475	327,850.7839	820,578.1674	39,409.21878	
	Best	5,174.145667	28,130.07444	45,063.73684	15,808.2439	10,972.8307	
	Std	15,563.51633	1,260,349.307	116,473.4587	333,452.1831	9,917.56265	
	p-value	0.000621601	0.004506605	0.255788656	0.054545455	–	

Table 8 Algorithm performance comparison on F1–F11 functions (CEC17, D30).

F	M	ISCSO	PGJA	SCA	WOA	IHO	
F1	Mean	292,017,921.2	54,602,176,926	11,944,943,682	4,401,883.079	1,937.38149	
	Worst	527,437,381	73,891,840,167	14,432,878,261	16,506,040.69	4,163.047573	
	Best	1,157,315.861	34,732,236,903	10,080,370,622	1,176,371.352	571.6410095	
	Std	242,841,632.6	14,505,437,048	1,445,590,005	5,148,145.471	1,194.963304	
	p-value	0.0001554	0.0001554	0.0001554	0.0001554	–	
F3	Mean	6,592.875554	193,862.9193	36,946.78344	152,112.0146	972.2301397	
	Worst	16,280.07041	257,517.6835	48,791.52284	231,347.5323	1,456.793182	
	Best	2,069.102652	123,307.0256	23,183.20465	69,120.91318	588.2283329	
	Std	4,394.691599	44,226.80563	8,873.788623	58,900.29004	358.3808352	
	p-value	0.0001554	0.0001554	0.0001554	0.0001554	–	
F4	Mean	521.2871451	15,390.90328	1,519.778972	531.0941842	514.1838143	
	Worst	540.3761612	29,569.24634	1,768.512026	594.2612394	521.9156609	
	Best	471.5028443	8,475.816002	1,189.835173	478.7331691	490.578135	
	Std	29.30041845	7,081.799406	201.2803899	32.32963674	9.947761249	
	p-value	0.104895105	0.0001554	0.0001554	0.104895105	–	
F5	Mean	729.7771593	856.9469445	778.2580037	774.3609374	728.8753933	
	Worst	757.6973487	925.3544808	800.9015842	831.9468905	746.7491739	
	Best	690.1336578	773.3180969	757.13216	707.341441	699.9856953	
	Std	23.95054976	44.58029759	14.33161208	36.40504277	21.27144608	
	p-value	0.441802642	0.0001554	0.000621601	0.028127428	–	
F6	Mean	646.6319023	683.4511262	648.9364883	669.6568011	651.280708	
	Worst	653.1660693	700.356083	658.1752722	688.2936903	657.8615813	
	Best	639.4347778	666.4831652	643.2417739	653.5274391	649.0870836	
	Std	4.482597405	14.64303146	5.39966958	11.56909408	4.061803166	
	p-value	0.001864802	0.0001554	0.028127428	0.01041181	–	
F7	Mean	1,196.584529	1,403.412487	1,110.917413	1,235.142759	1,054.778105	
	Worst	1,260.347902	1,464.117048	1,161.628717	1,394.803133	1,064.188065	
	Best	1,111.789344	1,325.532085	1,063.090669	1,109.529037	953.8814871	
	Std	52.59169901	46.64549253	40.56630954	116.5113486	90.23290204	
	p-value	0.006993007	0.0001554	0.441802642	0.037917638	–	
F8	Mean	973.5104573	1,129.481363	1,046.330124	1,004.69169	922.5286381	
	Worst	1,027.853338	1,189.891676	1,075.09119	1,058.843414	962.1857079	
	Best	924.521515	1,076.20449	1,034.548231	936.6616101	916.8633424	
	Std	38.7982171	35.46911457	13.6491679	39.99126557	16.02387599	
	p-value	0.194871795	0.0001554	0.0001554	0.001864802	–	
F9	Mean	5,222.372804	9,877.964111	5,187.794745	8,177.540975	4,183.031751	
	Worst	5,867.601194	13,102.14064	6,550.236105	11,305.23666	4,887.721362	
	Best	4,577.853746	6,294.855092	4,317.271029	5,749.647385	3,070.88618	
	Std	479.7332628	2,078.316099	730.1754764	1,743.327276	646.2343344	
	p-value	0.006993007	0.0001554	0.028127428	0.0001554	–	
F10	Mean	4,900.446449	8,769.033832	8,144.736264	6,108.605085	4,761.012918	
	Worst	5,633.035287	9,289.762328	8,465.174251	7,519.041975	5,196.529638	
	Best	3,856.79703	7,833.452431	7,802.535599	5,571.76996	4,078.291982	
	Std	563.0801194	456.5019826	225.5531433	598.497455	351.7157137	
	p-value	0.573737374	0.0001554	0.0001554	0.0001554	–	
F11	Mean	1,235.856792	16,272.53715	2,242.839168	1,485.187865	1,164.529094	
	Worst	1,266.464581	28,968.65435	4,347.208243	1,749.053725	1,165.529094	
	Best	1,198.065961	5,315.71357	1,646.069466	1,346.271373	1,161.877026	
	Std	22.800785	8,626.178484	871.5621147	128.2040341	35.72963828	
	p-value	0.064957265	0.0001554	0.0001554	0.0001554	–	

Table 9 Algorithm performance comparison on F12–F21 functions (CEC17, D30).

F	M	ISCSO	PGJA	SCA	WOA	IHO	
F12	Mean	3,893,735.818	10,762,991,447	1,204,150,091	53,544,181.27	2,739,363.429	
	Worst	6,136,774.139	12,631,613,989	1,558,108,812	106,630,291.9	5,338,630.393	
	Best	927,323.8437	8,704,282,079	893,265,690.2	21,095,292.26	649,972.9927	
	Std	1,837,315.004	1,417,182,823	245,946,936.1	31,721,212.59	1,708,162.482	
	p-value	0.13038073	0.0001554	0.0001554	0.0001554	–	
F13	Mean	66,848.22803	7,108,305,222	278,159,803.3	236,060.1539	55,020.19703	
	Worst	122,008.7687	16,620,595,721	576,404,881.3	661,885.213	88,225.81303	
	Best	32,169.76374	211,349,141	9,532,874.93	51,304.33087	21,968.71571	
	Std	26,916.26907	4,982,950,718	170,009,905	188,910.7041	22,061.70522	
	p-value	0.573737374	0.0001554	0.0001554	0.001864802	–	
F14	Mean	28,109.51102	7,994,248.012	115,088.8492	1,518,712.397	28,382.08989	
	Worst	50,614.75129	15,915,233.21	258,163.8981	4,095,454.8	58,915.79243	
	Best	4,903.370735	110,294.4551	30,832.44457	53,646.85126	3,362.728206	
	Std	18,815.74363	6,015,214.103	69,909.87692	1,487,251.065	20,130.70307	
	p-value	0.959129759	0.0001554	0.001864802	0.0003108	–	
F15	Mean	16,980.33092	823,753,944.9	14,554,206.93	61,766.98681	9,130.539813	
	Worst	50,830.10537	2,284,609,344	50,725,595.6	138,050.7618	11,300.42731	
	Best	5,531.727914	151,702,816.9	841,043.1355	18,477.97716	3,632.767213	
	Std	14,399.81892	666,426,828.7	16,299,656.58	44,554.8301	2,908.626101	
	p-value	0.441802642	0.0001554	0.0001554	0.0001554	–	
F16	Mean	2,999.107543	5,022.043056	3,532.109544	3,548.532546	2,885.769671	
	Worst	3,767.154767	6,043.114712	3,730.446138	3,989.026258	3,084.016852	
	Best	2,067.574106	3,751.13632	3,292.79637	2,845.474297	2,336.795091	
	Std	497.3112671	781.4674299	145.5982883	425.728683	353.9876677	
	p-value	0.505361305	0.0001554	0.0001554	0.006993007	–	
F17	Mean	2,442.655259	3,352.432549	2,410.73059	2,633.775657	2,111.266319	
	Worst	2,816.070687	3,789.297967	2,691.834453	3,022.684126	2,121.266319	
	Best	2,048.597595	2,971.061296	2,114.401126	2,199.598566	2,031.212632	
	Std	314.681462	342.3736545	181.0819424	282.6577852	152.0067553	
	p-value	0.160528361	0.0001554	0.04988345	0.004662005	–	
F18	Mean	80,356.95534	58,115,817.64	2,349,346.867	1,753,869.604	115,769.8552	
	Worst	126,927.4433	157,501,238.6	3,443,059.504	6,040,101.907	217,638.707	
	Best	40,390.9835	9,844,360.306	971,575.9576	131,232.1066	41,490.61988	
	Std	29,850.6702	48,382,069.71	775,955.834	1,875,590.054	55,170.35594	
	p-value	0.160528361	0.0001554	0.0001554	0.000621601	–	
F19	Mean	246,965.075	518,728,506.7	24,821,409.49	3,470,629.376	70,275.40673	
	Worst	446,145.1839	1,484,266,735	51,959,847.63	6,560,155.248	161,706.0163	
	Best	2,954.224537	32,366,446.67	11,553,243.37	1,406,647.559	4,935.709261	
	Std	173,052.6664	468,853,978.5	14,178,050.79	2,052,430.979	63,335.7412	
	p-value	0.064957265	0.0001554	0.0001554	0.0001554	–	
F20	Mean	2,567.281207	3,159.267152	2,579.284812	2,728.096849	2,436.261628	
	Worst	2,781.508133	3,392.528895	2,756.405947	3,059.465485	2,593.002205	
	Best	2,302.34815	2,729.407894	2,411.55585	2,387.230722	2,362.370376	
	Std	192.4560571	202.4511233	112.0988851	235.4730425	78.45269469	
	p-value	0.278632479	0.0001554	0.01041181	0.006993007	–	
F21	Mean	2,572.51916	2,694.186289	2,554.434603	2,559.012129	2,438.717795	
	Worst	2,642.311429	2,838.229827	2,576.84909	2,676.467007	2,438.717795	
	Best	2,509.890835	2,623.942895	2,531.639516	2,458.412602	2,438.717795	
	Std	51.14666156	76.2712779	18.39540383	62.17989791	0	
	p-value	0.13038073	0.0001554	0.064957265	0.328205128	–	

Table 10 Algorithm performance comparison on F22–F30 functions (CEC17, D30).

F	M	ISCSO	PGJA	SCA	WOA	IHO	
F22	Mean	3,094.1814	9,092.11298	8,471.509827	7,072.896845	3,686.286016	
	Worst	7,871.646366	10,228.67584	10,015.87024	9,962.733388	7,118.750543	
	Best	2,307.894963	5,853.17166	4,663.290857	4,371.685441	2,300.003441	
	Std	1,933.356211	1,588.089287	2,066.028385	1,648.110519	2,037.23275	
	p-value	0.573737374	0.000621601	0.001864802	0.01041181	–	
F23	Mean	3,012.683212	3,246.422092	2,981.313836	3,115.234801	2,874.494647	
	Worst	3,161.106827	3,468.868344	3,004.324878	3,212.76467	2,949.559422	
	Best	2,880.171663	3,044.450811	2,945.840725	2,873.263341	2,792.294751	
	Std	91.75925869	159.0001986	19.96224234	106.2430543	57.03912532	
	p-value	0.004662005	0.0001554	0.0003108	0.001864802	–	
F24	Mean	3,262.17368	3,454.817034	3,164.941193	3,207.096879	3,029.078137	
	Worst	3,334.903778	3,567.955707	3,196.220456	3,401.250435	3,035.370079	
	Best	3,100.531143	3,304.138491	3,131.798738	3,009.651887	2,957.550837	
	Std	73.33435879	87.66140419	21.66106942	130.6393863	51.63234843	
	p-value	0.001864802	0.0001554	0.006993007	0.037917638	–	
F25	Mean	2,917.66177	5,619.257499	3,187.527743	2,947.500442	2,945.519632	
	Worst	3,052.510397	7,394.318822	3,394.763524	2,986.311453	2,962.684802	
	Best	2,885.323924	4,268.59773	3,099.524355	2,908.564728	2,908.071022	
	Std	55.80294739	1,154.923286	92.62697519	26.92310133	17.61016888	
	p-value	0.01041181	0.0001554	0.0001554	0.194871795	–	
F26	Mean	7,042.629065	10,854.39371	6,809.200766	7,331.584475	6,665.625262	
	Worst	8,751.640492	12,973.2633	7,810.991274	8,226.531488	7,738.319778	
	Best	3,600.219462	8,612.54091	4,729.814685	6,641.802357	2,800.009595	
	Std	1,642.023335	1,574.961224	945.4346689	613.2100457	1,641.411716	
	p-value	0.278632479	0.0001554	0.278632479	0.04988345	–	
F27	Mean	3,365.180066	3,200.007275	3,381.463625	3,377.371312	3,356.756468	
	Worst	3,440.583453	3,200.007309	3,405.395808	3,497.983108	3,515.291845	
	Best	3,263.110573	3,200.007255	3,352.702104	3,257.317301	3,266.563595	
	Std	60.4009494	2.10E−05	18.26956401	66.49338878	88.45829752	
	p-value	0.720901321	1.55E-04	0.194871795	0.382284382	–	
F28	Mean	3,348.433888	3,300.007296	3,880.423941	3,323.201987	3,264.934756	
	Worst	3,458.499929	3,300.007344	4,458.943086	3,370.746706	3,276.515001	
	Best	3,221.564408	3,300.007255	3,673.597126	3,230.843037	3,213.320007	
	Std	96.78331233	3.61E−05	251.1540734	46.90082494	24.9135985	
	p-value	0.160528361	1.55E−04	0.0001554	0.01041181	–	
F29	Mean	4,304.267122	7,231.881019	4,617.214905	4,589.774637	4,257.69641	
	Worst	4,859.970391	9,130.545881	4,839.838089	5,104.602889	4,247.69641	
	Best	4,031.610678	5,485.274069	4,211.358315	3,991.744675	3,781.594522	
	Std	265.6988526	1,186.947621	232.9355674	428.6714343	380.6498471	
	p-value	0.328205128	0.0001554	0.001864802	0.04988345	–	
F30	Mean	679,632.1067	1,008,907,129	72,093,210.49	13,987,338.26	952,537.4116	
	Worst	2,177,700.418	2,020,243,537	96,868,364.84	20,825,059.16	1,482,768.199	
	Best	225,829.102	358,782,799.1	50,891,002.34	3,524,684.38	486,342.7661	
	Std	626,989.1644	617,846,992.6	18,720,181.93	6,035,567.003	414,447.396	
	p-value	0.037917638	0.0001554	0.0001554	0.0001554	–	

Table 11 Algorithm performance comparison on F1–F11 functions (CEC17, D50).

F	M	ISCSO	PGJA	IHO	
F1	Mean	2,314,099,692	1.09517E+11	136.840116	
	Worst	8,198,278,337	1.26697E+11	136.840116	
	Best	20,208,421.52	86,911,769,556	136.840116	
	Std	2,843,388,068	13,807,029,031	0	
	p-value	0.0001554	0.0001554	–	
F3	Mean	26,386.64688	397,051.7131	10,602.8233	
	Worst	38,357.0491	562,402.2137	11,451.01267	
	Best	14,650.02724	255,866.7868	8,058.255166	
	Std	8,544.87766	121,229.5124	1,570.541547	
	p-value	0.0001554	0.0001554	–	
F4	Mean	759.0707486	32,027.03949	537.2610501	
	Worst	1,120.646643	40,355.53108	537.2610501	
	Best	633.1780322	26,388.32262	537.2610501	
	Std	154.8346565	5,897.527889	0	
	p-value	0.0001554	0.0001554	–	
F5	Mean	869.955658	1,165.503128	879.0762527	
	Worst	911.0651916	1,247.942274	879.0762527	
	Best	840.2921017	1,057.015335	879.0762527	
	Std	22.29331944	55.22214722	1.22E−13	
	p-value	0.404195804	0.0001554	–	
F6	Mean	658.2406525	702.4874347	666.8369202	
	Worst	664.0948239	715.0579358	666.8369202	
	Best	646.8882899	691.6213642	666.8369202	
	Std	5.582102644	7.311618235	0	
	p-value	0.0001554	0.0001554	–	
F7	Mean	1,607.708753	2,594.14136	1,585.87234	
	Worst	1,660.602809	2,897.755392	1,598.024979	
	Best	1,534.491621	2,192.247545	1,500.803861	
	Std	45.20701757	235.3686815	34.37285611	
	p-value	0.255788656	0.0001554	–	
F8	Mean	1,174.289768	1,504.794049	1,151.606486	
	Worst	1,234.950511	1,590.087983	1,161.353849	
	Best	1,094.997975	1,391.408435	1,122.3644	
	Std	44.96484331	63.34009926	18.04860774	
	p-value	0.055633256	0.0001554	–	
F9	Mean	13,153.88226	35,528.15606	12,453.95075	
	Worst	14,722.03726	45,211.8866	12,453.95075	
	Best	11,462.79806	25,635.42303	12,453.95075	
	Std	981.4170759	7,669.769752	0	
	p-value	0.083139083	0.0001554	–	
F10	Mean	8,034.456962	14,540.15265	8,396.009456	
	Worst	8,385.917489	16,083.51594	8,396.009456	
	Best	7,623.952502	13,519.63647	8,396.009456	
	Std	291.8761871	968.1372401	1.94E−12	
	p-value	0.0001554	0.0001554	–	
F11	Mean	1,497.234556	39,836.45248	1,438.295597	
	Worst	2,039.311997	65,413.8759	1,438.295597	
	Best	1,291.997226	24,918.83344	1,438.295597	
	Std	274.3483854	16,165.17532	2.43E−13	
	p-value	0.404195804	0.0001554	–	

Table 12 Algorithm performance comparison on F12–F21 functions (CEC17, D50).

F	M	ISCSO	PGJA	IHO	
F12	Mean	18,565,838.19	72,253,248,327	15,172,139.37	
	Worst	61,470,238.03	1.14439E+11	36,806,291.76	
	Best	6,881,750.138	43,529,553,921	7,960,755.245	
	Std	17,744,990	20,649,104,729	13,352,888.75	
	p-value	0.211965812	0.0001554	–	
F13	Mean	115,673.4413	43,008,030,561	30,789.3711	
	Worst	187,464.0141	61,717,816,386	30,789.3711	
	Best	58,543.75843	21,754,994,609	30,789.3711	
	Std	41,702.14048	14,797,303,325	7.78E−12	
	p-value	0.0001554	0.0001554	–	
F14	Mean	150,776.8918	45,671,647.33	198,708.0026	
	Worst	397,490.9004	56,395,938.55	198,708.0026	
	Best	13,179.61048	30,666,512.55	198,708.0026	
	Std	144,330.1219	8,958,847.141	3.11E−11	
	p-value	0.404195804	0.0001554	–	
F15	Mean	32,526.42873	6,346,023,143	40,475.88833	
	Worst	45,655.50297	10,806,139,052	40,475.88833	
	Best	23,601.7272	3,702,532,382	40,475.88833	
	Std	7,919.675402	2,130,310,224	7.78E−12	
	p-value	0.083139083	0.0001554	–	
F16	Mean	4,088.56059	7,450.126177	3,881.020621	
	Worst	4,664.264423	9,426.512662	4,383.841565	
	Best	2,962.777752	5,728.470888	3,579.328055	
	Std	574.1646386	1,326.03294	416.3752989	
	p-value	0.225485625	0.0001554	–	
F17	Mean	3,544.74373	9,388.045439	3,156.434988	
	Worst	4,075.034571	22,133.00435	3,381.474316	
	Best	2,583.06957	4,581.519772	2,931.395659	
	Std	448.7058611	6,994.111059	240.5771616	
	p-value	0.032944833	0.0001554	–	
F18	Mean	603,500.6482	213,665,529	573,322.3636	
	Worst	967,792.8767	312,242,464.6	726,014.2105	
	Best	140,938.2612	60,582,287.99	115,246.823	
	Std	254,020.2798	92,960,961.08	282,730.3618	
	p-value	0.68982129	0.0001554	–	
F19	Mean	110,661.7469	2,894,965,576	383,732.6233	
	Worst	239,577.9614	5,331,080,081	466,838.3318	
	Best	18,915.58155	1,124,037,801	134,415.498	
	Std	66,649.18961	1,326,650,161	153,881.8705	
	p-value	0.001398601	0.0001554	–	
F20	Mean	3,490.99755	4,384.976119	3,034.259798	
	Worst	3,811.434398	4,834.057077	3,034.259798	
	Best	2,882.390684	3,817.158148	3,034.259798	
	Std	315.6157628	404.3570663	4.86E−13	
	p-value	0.005749806	0.0001554	–	
F21	Mean	2,852.755744	3,152.229775	2,711.42004	
	Worst	2,994.926201	3,266.921142	2,711.42004	
	Best	2,726.387748	2,981.223517	2,711.42004	
	Std	82.43139462	87.21052092	4.86E−13	
	p-value	0.0001554	0.0001554	–	

Table 13 Algorithm performance comparison on F22–F30 functions (CEC17, D50).

F	M	ISCSO	PGJA	IHO	
F22	Mean	10,097.24514	16,803.13079	9,732.499713	
	Worst	11,349.78656	17,464.97428	9,732.499713	
	Best	8,540.927621	16,043.0824	9,732.499713	
	Std	1,031.750704	526.810641	0	
	p-value	0.404195804	0.0001554	–	
F23	Mean	3,602.451297	4,045.506676	3,253.492207	
	Worst	3,871.667746	4,468.972728	3,253.492207	
	Best	3,411.751427	3,746.183096	3,253.492207	
	Std	145.7138778	249.9047003	4.86E−13	
	p-value	0.0001554	0.0001554	–	
F24	Mean	3,848.679083	4,276.830802	3,351.770618	
	Worst	4,013.647607	4,540.111469	3,351.770618	
	Best	3,668.632648	4,074.245681	3,351.770618	
	Std	139.8321169	190.8561342	9.72E−13	
	p-value	0.0001554	0.0001554	–	
F25	Mean	3,185.211572	15,882.78515	3,152.955715	
	Worst	3,596.612505	18,749.49117	3,236.749371	
	Best	2,968.714186	12,740.29072	3,140.985193	
	Std	221.423474	2,212.060556	33.85774996	
	p-value	0.365967366	0.0001554	–	
F26	Mean	8,777.798701	18,080.35163	10,606.71126	
	Worst	11,889.39314	19,558.16955	10,606.71126	
	Best	3,920.09441	17,277.78713	10,606.71126	
	Std	2,891.009547	733.2340026	0	
	p-value	0.404195804	0.0001554	–	
F27	Mean	4,399.218332	3,200.012303	3,832.989558	
	Worst	4,931.682129	3,200.012333	3,832.989558	
	Best	4,021.580808	3,200.012234	3,832.989558	
	Std	337.3443606	3.33E−05	0.00E+00	
	p-value	0.0001554	1.55E−04	–	
F28	Mean	3,492.179616	3,300.012256	3,682.983594	
	Worst	3,570.484076	3,300.012271	3,706.738994	
	Best	3,385.987202	3,300.012254	3,659.228193	
	Std	58.81338379	5.72E−06	2.54E+01	
	p-value	0.0001554	1.55E−04	–	
F29	Mean	5,476.629177	33,729.43285	7,268.356279	
	Worst	5,975.510304	87,404.64033	7,674.186487	
	Best	4,820.372827	15,559.58063	6,862.526071	
	Std	393.0630929	24,358.23358	433.8507415	
	p-value	0.0001554	0.0001554	–	
F30	Mean	15,148,162.29	5,828,428,279	73,112,987.03	
	Worst	26,376,873.21	8,004,252,232	73,112,987.03	
	Best	9,533,173.098	2,851,446,925	73,112,987.03	
	Std	5,037,618.121	1,892,099,398	0	
	p-value	0.0001554	0.0001554	–	

Table 14 Algorithm performance comparison on F1–F12 functions (CEC22, D10).

F	M	ISCSO	PGJA	SCA	WOA	IHO	
F1	Mean	300.0012578	21,338.68361	906.5997847	2,829.335784	300.0002745	
	Worst	300.001671	27,074.97094	1,746.427039	6,948.481906	300.0002745	
	Best	300.0006552	16,887.61786	559.3418705	852.7937575	300.0002745	
	Std	0.000360695	4,090.369186	515.2253636	2,021.677714	0	
	p-value	0.0001554	0.0001554	0.0001554	0.0001554	–	
F2	Mean	414.8410642	847.092998	440.7568793	415.7694813	400.0013254	
	Worst	470.7811823	1,178.825537	458.830874	470.9843296	400.0016179	
	Best	404.0431036	488.229109	424.6529838	400.5182491	400.000838	
	Std	22.72884453	259.0588808	13.33279631	22.72119917	0.00040364	
	p-value	0.0001554	0.0001554	0.0001554	0.0001554	–	
F3	Mean	610.4462434	645.518811	615.1519582	618.0276477	605.3286787	
	Worst	623.0026936	665.0160481	619.6103312	626.0617612	615.7245529	
	Best	604.5948396	622.1797317	611.9532154	609.7453232	603.8435538	
	Std	5.867560732	12.80481863	2.83449212	5.771840262	4.200567518	
	p-value	0.001087801	0.002486402	0.0001554	0.003574204	–	
F4	Mean	831.2168366	849.1022722	830.4806923	838.4271051	823.8789572	
	Worst	840.793418	873.8764745	837.1507328	852.7325001	823.8789572	
	Best	828.8537737	819.5417472	820.1171686	821.8615371	823.8789572	
	Std	4.082916517	17.38866659	6.492910164	12.08543922	1.22E−13	
	p-value	0.0001554	0.005749806	0.083139083	0.005749806	–	
F5	Mean	1,127.132015	1,698.560702	956.6005676	1,248.855047	1,058.149104	
	Worst	1,286.699153	2,774.698988	1,002.719039	1,523.89676	1,058.149104	
	Best	909.3275639	1,343.481464	931.7917725	931.0984207	962.3759894	
	Std	111.840472	484.3676296	28.26596108	217.3116082	50.24911778	
	p-value	0.083139083	0.0001554	0.0001554	0.083139083	–	
F6	Mean	2,215.112601	37,154,384.36	1,014,506.694	3,481.401386	1,875.065285	
	Worst	3,132.335475	153,147,587.9	1,948,446.737	7,289.229874	1,875.065285	
	Best	1,890.547321	481,459.4485	145,228.7776	1,937.256284	1,875.065285	
	Std	413.1683248	53,166,581.43	568,951.3022	1,855.875635	4.86E−13	
	p-value	0.0001554	0.0001554	0.0001554	0.0001554	–	
F7	Mean	2,056.321589	2,102.976131	2,046.161298	2,054.607512	2,030.758565	
	Worst	2,080.29621	2,168.781983	2,059.740497	2,110.136792	2,040.865563	
	Best	2,021.030765	2,058.764382	2,034.308974	2,017.281643	2,027.389565	
	Std	17.50209049	41.0124064	7.396737978	28.80401545	6.238174686	
	p-value	0.005749806	0.0001554	0.005749806	0.404195804	–	
F8	Mean	2,223.500609	2,237.046616	2,229.873114	2,229.785739	2,223.073336	
	Worst	2,226.232685	2,256.779258	2,232.604617	2,233.828848	2,223.073336	
	Best	2,220.34207	2,227.803163	2,225.677743	2,223.831819	2,223.073336	
	Std	2.022923903	11.22947611	2.172679836	3.781934734	0	
	p-value	0.404195804	0.0001554	0.0001554	0.0001554	–	
F9	Mean	2,529.284404	2,683.420966	2,540.361975	2,529.454456	2,529.324209	
	Worst	2,529.284425	2,806.680338	2,546.003124	2,529.934086	2,529.461261	
	Best	2,529.28439	2,560.456424	2,536.387211	2,529.289165	2,529.30463	
	Std	1.19E−05	89.14189293	3.368801013	0.26891251	0.05537734	
	p-value	1.55E−04	0.0001554	0.0001554	0.632012432	–	
F10	Mean	2,500.763953	2,633.763317	2,501.379955	2,531.058537	2,501.398645	
	Worst	2,501.588971	2,717.11002	2,501.692888	2,627.359992	2,501.398645	
	Best	2,500.349608	2,523.660092	2,501.153504	2,500.590289	2,501.398645	
	Std	0.380386122	81.77358517	0.163161169	56.25562509	0	
	p-value	0.005749806	0.0001554	0.404195804	0.083139083	–	
F11	Mean	2,619.110939	3,576.461769	2,757.536158	2,820.69916	2,912.204163	
	Worst	2,750.610263	4,177.032161	2,769.738377	3,000.115535	2,912.204163	
	Best	2,600.269189	2,798.869696	2,732.661559	2,600.325514	2,912.204163	
	Std	53.13375914	414.3148504	13.16798927	129.0200532	0	
	p-value	0.0001554	0.005749806	0.0001554	0.005749806	–	
F12	Mean	2,866.390826	2,887.361512	2,867.070403	2,874.959046	2,867.476747	
	Worst	2,869.03429	2,900.002364	2,868.105648	2,890.594673	2,872.827683	
	Best	2,863.874865	2,874.877797	2,866.174792	2,866.016365	2,865.221204	
	Std	1.615114155	10.20492632	0.585047573	9.10518868	2.162104734	
	p-value	0.71002331	0.0001554	0.31033411	0.146542347	–	

Table 15 Algorithm performance comparison on F1–F12 functions (CEC22, D20).

F	M	ISCSO	PGJA	SCA	WOA	IHO	
F1	Mean	488.8042873	52,158.81726	8,317.419385	1,029.367555	1,052.905909	
	Worst	804.0535642	84,288.25498	11,741.11935	1,678.114956	1,052.905909	
	Best	300.07585	26,667.36825	5,868.122615	516.056089	1,052.905909	
	Std	260.3916547	22,317.80721	1,776.063565	362.7102611	0	
	p-value	0.0001554	0.0001554	0.0001554	1	–	
F2	Mean	447.3834514	1,907.550058	622.4472336	481.4735839	455.319274	
	Worst	452.9074736	2,550.505744	687.3139401	565.7871322	455.319274	
	Best	444.9123181	1,096.912876	581.5163473	449.1374627	455.319274	
	Std	2.786319672	465.351248	33.45574714	43.72293039	0	
	p-value	0.0001554	0.0001554	0.0001554	0.404195804	–	
F3	Mean	632.2573257	679.1834401	631.9149429	655.9119578	645.3814982	
	Worst	646.6117706	694.2620618	638.395115	677.4479154	645.9315182	
	Best	620.9962574	669.0115305	621.8877361	638.107855	641.5313582	
	Std	7.98755262	8.185014182	5.471980267	14.26477887	1.555691481	
	p-value	0.0001554	0.0001554	0.0001554	0.68966589	–	
F4	Mean	881.5961829	967.5608162	928.9619702	914.1121341	872.6318082	
	Worst	910.440157	1,004.202627	941.6453729	1,008.016661	872.6318082	
	Best	854.7906913	945.4250628	917.6065107	859.7987221	872.6318082	
	Std	16.70932698	18.48610962	10.18169581	46.79015057	1.22E-13	
	p-value	0.088578089	0.0001554	0.0001554	0.005749806	–	
F5	Mean	2,555.926717	5,332.554773	1,986.496681	3,213.587228	2,049.345083	
	Worst	2,783.651934	8,907.179362	2,477.240028	5,200.323265	2,050.440548	
	Best	2,382.487316	2,914.518721	1,681.335774	2,488.711257	2,049.188588	
	Std	136.4568109	2,034.825769	276.9718404	868.5469296	0.442634661	
	p-value	0.0001554	0.0001554	0.090753691	0.0001554	–	
F6	Mean	4,075.697684	1,154,401,226	81,954,086.58	7,949.369402	2,677.887165	
	Worst	6,641.240621	2,583,141,493	149,857,095.6	20,786.25895	2,938.176238	
	Best	2,429.680752	291,954,563.6	20,375,879.8	2,195.946759	2,521.713722	
	Std	1,267.899961	719,990,911.7	41,721,914.5	7,663.163639	215.5398297	
	p-value	0.009479409	0.0001554	0.0001554	0.318414918	–	
F7	Mean	2,120.05609	2,260.937642	2,104.445225	2,164.418042	2,137.30073	
	Worst	2,187.496746	2,433.01941	2,120.083384	2,229.150558	2,137.30073	
	Best	2,071.509508	2,127.474371	2,081.140945	2,092.004779	2,136.636663	
	Std	35.41168296	97.98106173	16.0447879	49.37958283	1.2	
	p-value	0.005749806	0.005749806	0.0001554	0.083139083	–	
F8	Mean	2,240.200397	2,458.173285	2,245.755753	2,274.434031	2,230.445148	
	Worst	2,253.602792	2,713.626065	2,249.40696	2,376.551734	2,231.024571	
	Best	2,226.81159	2,239.791749	2,240.582754	2,232.883657	2,228.70688	
	Std	9.276854148	171.1019923	3.571372924	62.95429951	1.072882727	
	p-value	0.008391608	0.0001554	0.0001554	0.0001554		
F9	Mean	2,480.793405	2,961.059114	2,542.279169	2,486.065295	2,481.871841	
	Worst	2,480.804732	3,284.67978	2,575.337128	2,494.690517	2,481.871841	
	Best	2,480.783828	2,593.116114	2,520.524817	2,480.881462	2,481.871841	
	Std	0.006424571	205.6861281	21.20646064	5.360079074	0	
	p-value	0.0001554	0.0001554	0.0001554	0.083139083	–	
F10	Mean	2,570.431061	5,321.696779	2,518.082924	4,273.195864	2,500.832789	
	Worst	3,056.987303	7,211.498925	2,525.290057	5,396.578064	2,500.946918	
	Best	2,500.523244	2,899.972799	2,511.887519	2,500.919791	2,500.466397	
	Std	196.598616	1,951.02717	5.117671087	905.7961643	0.303401546	
	p-value	1	0.0001554	0.0001554	0.005749806	–	
F11	Mean	3,098.018288	7,622.174023	4,259.125479	3,217.164464	2,900.003623	
	Worst	3,363.530219	9,568.223698	4,682.75808	5,311.841662	2,900.003623	
	Best	2,604.4943	6,477.055237	3,659.227011	2,604.146351	2,900.003623	
	Std	338.3518427	970.2817973	419.1889007	860.8487609	4.86E−13	
	p-value	0.083139083	0.0001554	0.0001554	0.005749806	–	
F12	Mean	2,969.309989	2,900.004765	3,023.264466	3,030.207843	2,976.303321	
	Worst	3,002.925356	2,900.004782	3,049.894619	3,307.222416	2,976.303321	
	Best	2,953.774194	2,900.004753	3,011.539927	2,953.662188	2,976.303321	
	Std	16.43738863	1.09E−05	13.15477767	115.7593027	4.86E−13	
	p-value	0.083139083	0.0001554	0.0001554	0.404195804	–	

From Tables 5 to 15 shows four indicators, namely mean value, maximum value, minimum value and standard deviation. In addition, at the significance level of 5%, Wilcoxon rank sum tests are used to confirm whether IHO makes a significant contribution to other algorithms. “−” means “not applicable” which means that the best algorithm cannot be statistically compared to itself in the rank sum test.

The results of each evaluation are shown in Tables 5 to 15 for the total number of runs. The table also shows the p-values of the IHO and other algorithms, confirming the significant differences between the IHO proposed in this article and the other algorithms.

When dealing with CEC17, CEC22 benchmark functions, the convergence curve of IHO is compared with other classical algorithms, as shown in Figs. 8–12. Observing from this, it can be realized that many algorithms fall into the local solutions of most functions. For many functions, the IHO shows a high balance between the exploration and development phases.

Figure 8 Convergence curves of five algorithms in each benchmark functions (CEC17 D10).

Figure 9 Convergence curves of five algorithms in each benchmark functions (CEC17, D30).

Figure 10 Convergence curves of three algorithms in each benchmark functions (CEC17, D50).

Figure 11 Convergence curves of five algorithms in each benchmark functions (CEC22, D10).

Figure 12 Convergence curves of five algorithms in each benchmark functions (CEC22, D20).

The speed reducer problem (Eq. (17)) aims to minimize the total weight by adjusting parameters such as gear module, teeth number, and shaft length, while meeting stress, deformation, and gear ratio constraints. It involves mixed-integer nonlinear programming (e.g., integer teeth counts) and is a classic case in mechanical transmission optimization.

(17) minx0.7854x1x22(3.3333x32+14.9334x3−43.0934)−1.508x1(x62+x72)+7.4777(x63+x73)+0.7854(x4x62+x5x72)s.t.g1=27x1x22x3−1≤0,g2=397.5x1x22x32−1≤0,g3=1.93x43x2x64x3−1≤0,g4=1.93x53x2x74x3−1≤0,g5=(745x4x2x3)2+16.9×106110x63−1≤0,g6=(745x5x2x3)2+157.5×10685x73−1≤0,g7=x2x340−1≤0,g8=5x2x1−1≤0,g9=x112x2−1≤0,g10=1.5x6+1.9x4−1≤0,g11=1.1x7+1.9x5−1≤0,2.6≤x1≤3.6,0.7≤x2≤0.8,17≤x3≤28,7.3≤x4,x5≤8.3,2.9≤x6≤3.9,5.0≤x7≤5.4.

Gear train design problem (Eq. (18)) optimizes gear teeth numbers and modules (discrete variables) to minimize transmission ratio error, subject to geometric constraints (e.g., no undercutting) and material strength limits. It is widely used in multi-stage reducers and robotic joint design.

(18) minxf(x)=(16.931−x3x2x1x4)2s.t.12≤xi≤60,∀i∈{1,2,3,4}xi∈Z+,

Three-bar truss design problem (Eq. (19)) minimizes truss mass by adjusting cross-sectional areas of bars under stress, displacement, and stability constraints. Its nonlinear constraints and narrow feasible domain make it a benchmark for testing optimization algorithms.

(19) minx1,x2(22x1+x2)×100+1015∑k=13[gk2⋅H(gk)],H(gk)={1,gk > 00,gk ≤ 0s.t.g1:2(2x1+x2)2x12+2x1x2−2≤0g2:2x22x12+2x1x2−2≤0g3:22x2+x1−2≤00≤x1≤1,0≤x2≤1.

When dealing with three engineering optimization problems, IHO is compared with other classical algorithms, as shown in Table 16. From these observations, we can know that IHO performs better than other algorithms.

Table 16 Algorithm performance comparison on engineering problems.

Problem	Metrics	GJO	PGJA	IHO	
Speed reducer	Mean	3,002.3409	3,410.0096	2,983.1335	
	Worst	3,012.5920	4,127.4708	2,992.5352	
	Best	2,996.4256	3,189.4433	2,894.7444	
	Std	6.1953	402.2998	18.4221	
Gear train design	Mean	1.69×10−14	6.78×10−12	0	
	Worst	4.36×10−14	3.39×10−11	0	
	Best	2.19×10−16	0	0	
	Std	2.18×10−14	1.52×10−11	0	
Three bar truss design	Mean	263.8969	264.0344	263.8958	
	Worst	263.8976	264.2412	263.8958	
	Best	263.8961	263.9156	263.8958	
	Std	5.41×10−4	0.1242	3.65×10−5	

Hyperparameter sensitivity analysis

This subsection also provides a sensitivity analysis of IHO to the hyperparameter N. To examine the sensitivity of HO to hyperparameter N, the proposed algorithm is applied across varying values of N, specifically 20, 30, 50, and 100. These differing values of N are utilized to optimize the performance on benchmark functions F1 to F23. The optimization results are presented in Table 17. The sensitivity analysis of IHO to the hyperparameter N indicates that increasing the number of search agents enhances IHO’s capability in exploring the search space, thereby improving the performance of the proposed algorithm and reducing objective function values.

Table 17 Function values for different N (CEC05).

Fun	N=20	N=30	N=50	N=100	
F1	0.000000×100	0.000000×100	0.000000×100	0.000000×100	
F2	0.000000×100	0.000000×100	0.000000×100	0.000000×100	
F3	0.000000×100	0.000000×100	0.000000×100	0.000000×100	
F4	0.000000×100	0.000000×100	0.000000×100	0.000000×100	
F5	2.340199×10−2	1.225009×10−4	6.915110×10−3	4.712653×10−4	
F6	0.000000×100	0.000000×100	0.000000×100	0.000000×100	
F7	1.789226×10−5	3.078213×10−5	8.631090×10−5	8.970116×10−6	
F8	−1.256949×104	−1.256946×104	−1.256949×104	−1.256949×104	
F9	0.000000×100	0.000000×100	0.000000×100	0.000000×100	
F10	4.440892×10−16	4.440892×10−16	4.440892×10−16	4.440892×10−16	
F11	0.000000×100	0.000000×100	0.000000×100	0.000000×100	
F12	2.197844×10−4	4.980854×10−7	8.288517×10−9	5.816707×10−5	
F13	4.965446×10−31	1.359225×10−4	5.141361×10−5	4.638197×10−5	
F14	9.980038×10−1	9.980038×10−1	9.980038×10−1	9.980038×10−1	
F15	3.076108×10−4	3.074877×10−4	3.074866×10−4	3.074879×10−4	
F16	−1.031628×100	−1.031628×100	−1.031628×100	−1.031628×100	
F17	3.978874×10−1	3.978874×10−1	3.978874×10−1	3.978874×10−1	
F18	3.000000×100	3.000000×100	3.000000×100	3.000000×100	
F19	−3.862782×100	−3.862782×100	−3.862782×100	−3.862782×100	
F20	−3.321995×100	−3.321995×100	−3.321995×100	−3.321995×100	
F21	−1.015320×101	−1.015320×101	−1.015320×101	−1.015320×101	
F22	−1.040294×101	−1.040294×101	−1.040294×101	−1.040294×101	
F23	−1.053641×101	−1.053641×101	−1.053641×101	−1.053641×101	

Analysis of advantages and disadvantages of the improved algorithm

Through a series of standard benchmark functions, this study validates the performance of the improved IHO. The experimental results indicate that the improved algorithm outperforms the standard HO and other common metaheuristic algorithms in both convergence speed and solution accuracy. Notably, IHO demonstrated stronger global search capability on multi-modal functions. The evaluation results are presented in Tables 2–4, and Figs. 2, 5 displays the convergence curves of the three most effective algorithms in optimizing CEC05, CEC17, CEC22 and engineering optimization problems. The key findings are summarized as follows.

IHO consistently outperformed other algorithms in terms of convergence speed across a range of benchmark functions, achieving global optima in multiple cases, demonstrating superior convergence. Additionally, the improved algorithm provided higher solution accuracy compared to the standard HO and other metaheuristic algorithms, particularly in multi-modal optimization problems. Furthermore, IHO exhibited lower standard deviation, indicating greater stability in maintaining solution quality over multiple independent runs.

While IHO demonstrates substantial improvements over HO and other algorithms, some limitations remain, such as parameter sensitivity and the computational cost for large-scale problems. However, the overall performance of IHO shows that it is a promising optimization algorithm for solving complex optimization problems.

Conclusion

In this article, we introduced an improved version of the hippopotamus optimization algorithm by implementing three key modifications to enhance its performance. The first modification introduces a gradual reduction of the inertia weight, which helps balance the exploration and exploitation phases, allowing the algorithm to search the solution space more efficiently. The second modification involves adaptively adjusting the mutation rate based on the iteration count, improving the precision of the search in the later stages and reducing the risk of premature convergence. Finally, the predator escape phase has been refined with local boundary updates and position adjustments, resulting in significantly improved global convergence and a greater ability to avoid local optima.

The improved HO was benchmarked on a set of optimization problems, where its performance was compared against the original HO and several well-known metaheuristic algorithms. Results demonstrated that the improved HO consistently achieved superior solutions with higher accuracy, while effectively avoiding common issues such as getting trapped in local minima. The adaptive mechanisms incorporated into the algorithm improved its balance between exploration and exploitation, particularly in complex, high-dimensional search spaces.

Comparative experiments on functions (CEC05, CEC17 and CEC22), using metrics like mean, variance, median, p-value, convergence curves, and box plots, confirmed the enhanced performance of the modified HO over the original and other algorithms. These results highlight the effectiveness and robustness of the proposed improvements in addressing diverse optimization challenges.

In conclusion, the proposed enhancements to the HO algorithm have demonstrated their ability to significantly improve global optimization performance. Future work could explore further extensions of this algorithm, including multi-objective or binary versions, and investigate its application to a wider range of real-world optimization problems across various domains.

Supplemental Information

Supplemental Information 1 IHOA code.

Additional Information and Declarations

Competing Interests

Gang Sun is an employee of the Hunan Tobacco Workers Training Center.

Author Contributions

Shengyu Pei conceived and designed the experiments, performed the experiments, performed the computation work, prepared figures and/or tables, authored or reviewed drafts of the article, and approved the final draft.

Gang Sun performed the experiments, analyzed the data, prepared figures and/or tables, and approved the final draft.

Lang Tong performed the experiments, prepared figures and/or tables, and approved the final draft.

Data Availability

The following information was supplied regarding data availability:

The data are available in the Supplemental Files.

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
