# Peer review of "An improved hippopotamus optimization algorithm based on adaptive development and solution diversity enhancement"

_PeerJ Computer Science, doi:10.7717/peerj-cs.2901_

## Round 0.1 · original submission · Major Revisions

Dear Authors,

Thank you for the submission. The reviewers’ comments are now available. It is not suggested that your article be published in its current format. We do, however, advise you to revise the paper in light of the reviewers’ comments and concerns before resubmitting it.

Best wishes,

Reviewer 1 ·

Basic reporting

no comment

Experimental design

no comment

Validity of the findings

no comment

Additional comments

Q.1. How does the logistic chaos map initialization strategy affect the convergence of the HOA algorithm in multi-vertex search spaces?
Q.2. How can Gaussian mutation combined with chaotic perturbation maintain diversity in the population without reducing the efficiency of the exploitation and exploration processes in the HOA algorithm?
Q.3. There is other improved HOA as https://link.springer.com/chapter/10.1007/978-981-97-5578-3_21, what is the difference?
Q.4. The review of literature should cover more recent algorithms. Many recent works are missing in the review of the literature.
Q. 5. Benchmarking the algorithm must be tested on a more challenging standard test suite for truss optimization. All or some of the examples from the test suite below need to be tested to verify the efficiency of the proposed method: A standard benchmarking suite for structural optimization algorithms: ISCSO 2016–2022, Structures 58, 105409.
Q.6. CEC 2015-2024 should be presented.

Reviewer 2 ·

Basic reporting

The manuscript presents a new approach to enhancing the Hippopotamus Optimization Algorithm (HOA) through adaptive development and solution diversity mechanisms. The proposed improvements, such as chaotic map initialization and Gaussian mutation-based diversity enhancement, effectively address common limitations of metaheuristic algorithms, including premature convergence and poor scalability in high-dimensional spaces. These contributions are demonstrated through extensive benchmarking on 23 standard functions, with results showing superior convergence speed and solution accuracy compared to other established algorithms like PSO and CMA-ES.The manuscript provides a solid foundation in terms of literature references and field context. It reviews key optimization algorithms, including evolutionary, swarm intelligence-based, and metaheuristic approaches, highlighting their strengths and limitations. Additionally, it positions the Hippopotamus Optimization Algorithm (HOA) within this framework, emphasizing its unique behaviors and contributions. However, the review of related works could be expanded to include more recent advancements in metaheuristic optimization, particularly those addressing similar challenges, such as premature convergence and solution diversity. For example, recent studies on hybrid algorithms or adaptive strategies in swarm intelligence could provide a more comprehensive background. Moreover, while the manuscript discusses benchmark functions, it lacks explicit justification for their selection and how they represent challenges encountered in real-world applications.

Experimental design

The manuscript presents original research that aligns well with the journal's aims, focusing on improving the Hippopotamus Optimization Algorithm (HOA) with adaptive and diversity-enhancing strategies. The research question is clearly defined and addresses a meaningful gap in optimization methodologies, particularly the challenges of premature convergence and low diversity in metaheuristic algorithms. The investigation is rigorous, with comprehensive testing on benchmark functions, comparisons to state-of-the-art algorithms, and robust statistical analyses, ensuring the reliability of the results.

Validity of the findings

The manuscript demonstrates significant potential in its impact and novelty, but these aspects are not fully articulated. While the proposed improvements to the Hippopotamus Optimization Algorithm (HOA) are innovative, the broader implications of these advancements, especially in real-world applications, are not discussed in detail. Expanding on how this work contributes to or advances the field beyond benchmark tests would strengthen its impact.

The rationale for the study is well-stated, and the findings are connected to a meaningful research gap in optimization algorithms.

The underlying data presented in the manuscript appear statistically robust, sound, and well-controlled. The use of established statistical metrics and benchmark functions supports the validity of the results.

The conclusions are well-articulated and align with the original research question, staying appropriately limited to the supporting results.

·

Basic reporting

In this paper, an improved version of the hippopotamus optimization algorithm by implementing three key modifications to enhance its performance is introduced.

The authors present a very nice introduction, and a very nice definition of the problem and other proposed optimization algorithms reported in the literature. However, it is not ready to be accepted yet in Peerj journal due to the following comments.

1) In related work section, the authors ignored one of the most recent algorithms in the human behavior and social dynamics direction. It is Gaining Sharing Knowledge-based Algorithm GSK)

Thus, it must be added to this direction.

Experimental design

1) Actually, using out-of-date classical problems (23) are not enough for comparison. Thus, the author must evaluate the proposed algorithm on all CEC2017 benchmark problems on D=10,30,50 and 100. The number of runs is 51 for each problem in each dimension, the total function evaluations are 100000,300000,500000 and 1000000 for D=10,30,50,100.

Note that: F2 must be removed as it is unstable problem and has been removed from CEC2017 competition.

Note that: the current out-of-date benchmark problems must be removed.


3) The convergence figures of all compared algorithms on the benchmark problems in all dimensions (10,30, 50 and 100) must be provided.

Validity of the findings

1) Besides, it is unfair comparisons as the authors compared advanced version of IHOA with basic PSO, GA and CSA algorithms. Thus, you have to experimentally investigate the performance of the proposed algorithm using CEC 2017 (for unconstrained) with dimensions (10, 30, 50 and 100) and make a comparison with up to date algorithms.
2) The appropriate statistical analysis must be done using one of the non-parametric statistical tests such as Wilcoxon Signed Rank Test and Fridman. i.e. Statistical tests must be included:

Note that: You can add more recent ADVANCED algorithms for comparison.

Reviewer 4 ·

Basic reporting

The authors propose an improved hippopotamus optimization algorithm to address the limitations of the traditional hippopotamus optimization algorithm in terms of convergence performance and solution diversity in complex high-dimensional problems.

The article is well written. The contribution is clear. The method is formally presented. There is a relevant set of case studies with comparative analyses.

Experimental design

The method is well presented and well discussed.

Validity of the findings

There are well-conducted and discussed case studies.

---

## Round 0.2 · accepted · Accept

Dear Authors,

Thank you for addressing the reviewers comments. Your paper seems sufficiently improved and ready for publication.

Best wishes,

Reviewer 1 ·

Basic reporting

no comment

Experimental design

no comment

Validity of the findings

no comment

Additional comments

no comment